# Towards Unifying Behavioral and Response Diversity for Open-ended Learning in Zero-sum Games

**Xiangyu Liu**[1], **Hangtian Jia**[2], **Ying Wen**[1]*, **Yujing Hu**[2],
**Yingfeng Chen**[2], **Changjie Fan**[2], **Zhipeng Hu**[2], **Yaodong Yang**[3]
[1]Shanghai Jiao Tong University, [2]Netease Fuxi AI Lab, [3]Institute for AI, Peking University
liuxiangyu999@sjtu.edu.cn, ying.wen@sjtu.edu.cn, yaodong.yang@pku.edu.cn

## Abstract

Measuring and promoting policy diversity is critical for solving games with strong non-transitive dynamics where strategic cycles exist, and there is no consistent winner (e.g., Rock-Paper-Scissors). With that in mind, maintaining a pool of diverse policies via open-ended learning is an attractive solution, which can generate auto-curricula to avoid being exploited. However, in conventional open-ended learning algorithms, there are no widely accepted definitions for diversity, making it hard to construct and evaluate the diverse policies. In this work, we summarize previous concepts of diversity and work towards offering a unified measure of diversity in multi-agent open-ended learning to include all elements in Markov games, based on both *Behavioral Diversity (BD)* and *Response Diversity (RD)*. At the trajectory distribution level, we re-define BD in the state-action space as the discrepancies of occupancy measures. For the reward dynamics, we propose RD to characterize diversity through the responses of policies when encountering different opponents. We also show that many current diversity measures fall in one of the categories of BD or RD but not both. With this unified diversity measure, we design the corresponding diversity-promoting objective and population effectivity when seeking the best responses in open-ended learning. We validate our methods in both relatively simple games like matrix game, non-transitive mixture model, and the complex *Google Research Football* environment. The population found by our methods reveals the lowest exploitability, highest population effectivity in matrix game and non-transitive mixture model, as well as the largest goal difference when interacting with opponents of various levels in *Google Research Football*.

## 1 Introduction

Many zero-sum games have a strong non-transitive component [2, 4] in the policy space, and thus each player must acquire a diverse set of winning strategies to achieve high unexploitability [36], which has been widely validated by recent studies of constructing AIs with superhuman performance in sophisticated tasks, such as StarCraft [30, 28] and DOTA2 [25, 39]. The non-transitivity in games means there is not a dominating strategy and the set of strategies form a cycle (e.g., the endless cycles among Rock, Paper and Scissors). It is the presence of this special structure in games that requires players to maintain a diverse set of policies. Otherwise, we only need to seek the strongest one. Formally, the necessity of diversity for zero-sum games lies in three ways: (1) policy evaluation: with the presence of non-transitivity, one cannot justify the strength or weakness of a strategy through the outcome of the interaction with a single type of opponent; (2) avoiding being exploited [24]: since in non-transitive games a single strategy can be always beaten by another one, a diverse set of strategies

---

*Correspondence to Ying Wen <ying.wen@sjtu.edu.cn>. Code available at https://github.com/sjtu-marl/bd_rd_psro

allows players to make corresponding responses when encountering different opponents; (3) training adaptable strategies [32]: a diverse set of training opponents helps gradually eliminate the weakness of a strategy, which can adapt to a wide range of opponents with very few interactions at test time.

The open-ended learning framework is a promising direction towards inducing a population of distinct policies in zero-sum games via auto-curricula. Although various open-ended algorithms have been proposed to derive diverse strategies [2, 24, 6, 22], there are no consistent definitions for diversity. One of the most intuitive principles to characterize diversity is to build metrics over the trajectory or state-action distribution [6, 20]. However, this perspective only focuses on the policy behaviors and ignores the reward attributes inherited from the Markov decision process. We argue that this is not reasonable since sometimes a slight difference in the policy can result in a huge difference in the final reward like, in maze. Contrary to this, another line of works builds the diversity measure over empirical payoffs [24, 2, 10], thus revealing the underlying diverse behaviors of a strategy through the responses when encountering distinct opponents.

In this work, based on all previous diversity concepts, we work towards offering a unified view for diversity in an open-ended learning framework by combining both the behavioral attribute and the response attribute of a strategy. The behavioral diversity is formulated through the occupancy measure, which is an equivalent representation of a policy. We hypothesize that the diversity in policy behaviors should be revealed by differences in the state-action distribution, and we use a general divergence family $f$-divergence to indicate the novelty of a new policy. On the other hand, gamescape [2] has been proposed to represent the response capacity of a population of strategies. Based on gamescape, we formulate a new geometric perspective to treat the response diversity by considering the distance to the gamescape.

To summarize, in this paper, we provide the following contributions:

- We formulate the concept of *behavioral diversity* in the state-action space as the discrepancies of occupancy measures and analyze the optimization methods in both normal-form games and general Markov games.

- We provide a new geometric perspective on the *response diversity* as a form of Euclidean projection onto the convex hull of the meta-game to enlarge the gamescape directly and propose the optimization lower bound for practical implementation.

- We analyze the limitation of exploitability as the evaluation metric and introduce a new metric with theoretical soundness called *population effectivity*, which is a fairer way to represent the effectiveness of a population than exploitability [16].

## 2 Preliminaries

### 2.1 Markov Games

The extension of Markov decision processes (MDPs) with more than one agents is commonly modelled as Markov games [18]. A Markov game with $N$ agents is defined by a tuple $< N, \mathcal{S}, \{\mathcal{A}_i\}_{i=1}^N, P, \{r_i\}_{i=1}^N, \eta, \gamma >$, where $\mathcal{S}$ denotes the state space and $\mathcal{A}_i$ is the action space for agent $i$. The function $P$ controls the state transitions by the current state and one action from each agent: $P : \mathcal{S} \times \mathcal{A}_1 \times \cdots \times \mathcal{A}_N \to \mathcal{P}(\mathcal{S})$, where $\mathcal{P}(\mathcal{S})$ denotes the set of probability distributions over the state space $\mathcal{S}$. Given the current state $s_t$ and the joint action $(a_1, \ldots, a_N)$, the transition probability to $s_{t+1}$ is given by $P(s_{t+1}|s_t, a_1, \ldots, a_N)$. The initial state distribution is given by $\eta : \mathcal{S} \to [0, 1]$. Each agent $i$ also has an associated reward function $r_i : \mathcal{S} \times \mathcal{A}_i \times \cdots \times \mathcal{A}_N \to \mathbb{R}$. Each agent's goal is to maximize the $\gamma$-discounted expected return $R_i = \mathbb{E}[\sum_{t=0}^\infty \gamma^t r_i(s_t, a_i^t, a_{-i}^t)]$, where $-i$ is a compact representation of all complementary agents of $i$. Specifically, for zero-sum games, the rewards satisfy that $\sum_{i=1}^N r_i(s, \mathbf{a}) = 0$, and players need to behave competitively to achieve higher rewards.

In multi-agent reinforcement learning (MARL) [37], each agent is equipped with a policy $\pi_i : \mathcal{S} \times \mathcal{A}_i \to [0, 1]$ and the joint policy is defined by $\boldsymbol{\pi}(\mathbf{a}|s) = \Pi_{i=1}^N \pi_i(a_i|s)$. In single-agent reinforcement learning, *occupancy measure* is a principled way to characterize a policy, which indicates how a policy covers the state-action space. Inspired by the definition from the single-agent setting, we define the joint occupancy measure in MARL induced by the joint policy $\boldsymbol{\pi}(\mathbf{a}|s)$ as:

Table 1: Comparisons of Different Algorithms.

| Method | Tool for Diversity | BD | RD | Game Type |
|---|---|---|---|---|
| DvD | Determinant | ✓ | × | Single-agent |
| PSRO$_N$ | None | × | × | n-player general-sum game |
| PSRO$_{rN}$ | $L_{1,1}$ norm | × | ✓ | 2-player zero-sum game |
| DPP-PSRO | Determinantal point process | × | ✓ | 2-player general-sum game |
| Our Methods | Occupancy measure & convex hull | ✓ | ✓ | n-player general-sum game |

**Definition 1.** *(Joint Occupancy Measure in MARL) Let $\rho_{\boldsymbol{\pi}}(s) : \mathcal{S} \to \mathbb{R}$ denote the normalized distribution of state visitation by following the joint policy $\boldsymbol{\pi} = (\pi_i, \pi_{-i})$ in the environment:*

$$\rho_{\boldsymbol{\pi}}(s) = (1 - \gamma) \sum_{t=0}^{\infty} \gamma^t P(s_t = s | \boldsymbol{\pi}) . \tag{1}$$

*Then the distribution of state-action pairs $\rho_{\boldsymbol{\pi}}(s, \mathbf{a}) = \rho_{\boldsymbol{\pi}}(s)\boldsymbol{\pi}(\mathbf{a}|s)$ is called occupancy measure of the joint policy $\boldsymbol{\pi}$.*

## 2.2 Policy Space Response Oracle

Adapted from double oracle [23], policy space response oracle (PSRO) [16, 5] has been serving as a powerful tool to solve the nash equilibrium (NE) in zero-sum games. In PSRO, each player maintains a pool of policies, say $\mathfrak{P}_i = \{\pi_i^1, \ldots, \pi_i^M\}$ for player $i$ and $\mathfrak{P}_{-i} = \{\pi_{-i}^1, \ldots, \pi_{-i}^N\}$ for player $-i$. The so-called meta game $\mathbf{A}_{\mathfrak{P}_i \times \mathfrak{P}_{-i}}$ has its $(k, j)$ entry as $\phi_i(\pi_i^k, \pi_{-i}^j)$, where the function $\phi_i$ encapsulates the reward outcome for player $i$ like the winning rate or expected return. When player $i$ adds a new policy $\pi_i^{M+1}$, it will compute the best response to the mixture of its opponents:

$$\text{Br}(\mathfrak{P}_{-i}) = \max_{\pi_i^{M+1}} \sum_j \sigma_{-i}^j \mathbb{E}_{\pi_i^{M+1}, \pi_{-i}^j}[r_i(s, \mathbf{a})].$$

where $\boldsymbol{\sigma} = (\sigma_i, \sigma_{-i})$ is a distribution over policies in $\mathfrak{P}_i$ and $\mathfrak{P}_{-i}$, which is usually a NE of $\mathbf{A}_{\mathfrak{P}_i \times \mathfrak{P}_{-i}}$.

The empirical gamescape is introduced by [2] to represent the expressiveness of a population $\mathfrak{P}_i$ in the reward outcome level given the opponent population $\mathfrak{P}_{-i}$:

**Definition 2.** *Given population $\mathfrak{P}_i$ and $\mathfrak{P}_{-i}$ with evaluation matrix $\mathbf{A}_{\mathfrak{P}_i \times \mathfrak{P}_{-i}}$, the corresponding empirical gamescape (EGS) for $\mathfrak{P}_i$ is defined as*

$$\mathcal{G}_{\mathfrak{P}_i | \mathfrak{P}_{-i}} := \{convex \ mixtures \ of \ rows \ of \ \mathbf{A}_{\mathfrak{P}_i \times \mathfrak{P}_{-i}}\}.$$

## 2.3 Existing Diversity Measures

As the metric to measure the differences between models, diversity is an important topic in many fields of machine learning, including generative modelling [6], latent variable models [33], and robotics [1]. Specifically, in reinforcement learning (RL), diversity is a useful tool for learning transferable skills [7], boosting explorations [27], or collecting near-optimal policies that are distinct in a meaningful way. Despite the importance of diversity, as shown in Table 1, there has not been a consistent definition of diversity for RL, and various diversity concepts are used. [22] investigated **behavioral diversity** in multi-agent reinforcement learning through *expected action variation*, which is modeled as the average total variation distance of two action distributions under certain sampled states. Considering the geometric perspective that the determinant of the kernel matrix represents the volume of a parallelepiped spanned by feature maps, DvD [26] proposed the concept of **population diversity** using the determinant of the kernel matrix composed by the behavioral embeddings by multiple policies. Thanks to the tools of empirical game theory analysis, diversity can be modeled from the perspective of the empirical game. **Effective diversity** [2] is formulated as the weighted $L_{1,1}$ norm of the empirical payoff matrix, which emphasizes what opponents a policy can win against. Also inspired by determinantal point process (DPP) [14, 38], [24] uses the **expected cardinality** to measure the diversity of a population.

# 3 A Unified Diversity Measure

Motivated by bisimulation metrics [8] to measure the similarity of two states in MDPs: $d\left(\mathbf{s}_i, \mathbf{s}_j\right) = \max_{\mathbf{a} \in \mathcal{A}}(1-c) \cdot \left|\mathcal{R}_{\mathbf{s}_i}^{\mathbf{a}} - \mathcal{R}_{\mathbf{s}_j}^{\mathbf{a}}\right| + c \cdot W_1\left(\mathcal{P}_{\mathbf{s}_i}^{\mathbf{a}}, \mathcal{P}_{\mathbf{s}_j}^{\mathbf{a}}; d\right)$ that considers both the immediate reward and the following transition dynamics, where $c \in [0, 1]$ and $W_1$ is the 1-Wasserstein distance, we want to build a metric to measure the similarity of two policies in a given Markov game through the task-specific reward attributes and the interaction between policy behaviors and transition dynamics. We will firstly model the interaction between policy behaviors and transition dynamics through the principled occupancy measure in MDPs, which encodes how a policy behaves in a given state and how the state will transit. On the reward side, the interaction responses with different opponents feature a policy, which can be used for common diversity measures like DPP [24] and rectified Nash [2].

## 3.1 Behavioral Diversity via Occupancy Measure Mismatching

One fundamental way to characterize a policy in MDPs is through the distribution of the state-action pairs $(s, a)$. Formally, we define the occupancy measure in multi-agent learning as the distribution of the joint state-action distribution. It has been shown that there is a one-to-one correspondence between the joint policy $\pi$ and the occupancy measure $\rho_\pi$.

**Proposition 1** (Theorem 2 of [29]). *If $\rho$ is a valid occupancy measure, then $\rho$ is the occupancy measure for $\pi_\rho(a \mid s) = \rho(s, a)/\sum_{a'} \rho\left(s, a'\right)$, and $\pi_\rho$ is the only policy whose occupancy measure is $\rho$.*

Usually, the policy $\pi$ is parameterized as a neural network, and tackling the policy in the parameter space is intractable. However, due to the one-to-one correspondence between the policy and occupancy measure, the occupancy measure $\rho_\pi$ serves as a unique and informative representation for the policy $\pi$. Therefore, we are justified in considering diversity from a perspective of the occupancy measure.

Next, we will consider how to promote diversity in the framework of policy space response oracle. Suppose after $t$ iterations of PSRO, the joint policy aggregated according the distribution of nash is $\boldsymbol{\pi_E} = (\pi_i, \pi_{E_{-i}})$. The occupancy measure is given by $\rho_{\boldsymbol{\pi_E}}$. For player $i$ in the $t+1$ iteration, it will seek the new policy $\pi_i'$, which can maximize the discrepancy between old $\rho_{\boldsymbol{\pi_E}}$ and $\rho_{\pi_i', \pi_{E_{-i}}}$.

$$\max_{\pi_i'} \text{Div}_{\text{occ}}(\pi_i') = D_f(\rho_{\pi_i', \pi_{E_{-i}}} || \rho_{\pi_i, \pi_{E_{-i}}}), \tag{2}$$

where we use the general $f$-divergence to measure the discrepancy of two distributions.

We firstly investigate the objective under the one-step game by giving the following theorem:

**Theorem 1.** *By assuming the game is a one-step game (normal-form games, functional-form games, etc.) and policies among players are independent, the behavioral diversity can be simplified by:*

$$D_f(\rho_{\pi_i', \pi_{E_{-i}}} || \rho_{\pi_i, \pi_{E_{-i}}}) = \mathbb{E}_{s_0 \sim \eta(s)}[D_f(\pi_i'(\cdot|s_0)||\pi_i(\cdot|s_0))], \tag{3}$$

*where $\eta(s)$ is the initial state distribution.*

*Proof.* See Appendix A.1. □

For more general Markov games, computing the exact occupancy measure is intractable. However, notice that we are maximizing a $f$-divergence objective of occupancy measures, while occupancy measure matching algorithms in imitation learning try to minimize the same objective [13, 11, 9]. Leveraging the powerful tool from occupancy measure matching, we here propose an approximate method to maximize $\text{Div}_{\text{occ}}$.

**Prediction Error for Approximate Optimization.** Inspired by random expert distillation [31], a neural network $f_{\hat{\theta}}(s, \mathbf{a})$ is trained to fit a randomly initialized fixed network $f_\theta(s, \mathbf{a})$ on the dataset of state-action pair $(s, \mathbf{a}) \sim \rho_{\boldsymbol{\pi_E}}$. Then we can assign an intrinsic reward $r_i^{int}(s, \mathbf{a}) = ||f_{\hat{\theta}}(s, \mathbf{a}) - f_\theta(s, \mathbf{a})||$ to the player, which will encourage the agent to visit the state-action with large prediction errors, thus pushing occupancy measure of the new policy to be different from the old one.

**Alternative Solutions.** There are also many other practical occupancy measure matching algorithms. One popular paradigm is learning a discriminator $D(s, \mathbf{a})$ to classify the state-action pair $(s, \mathbf{a})$ from the distribution $\rho_{\pi_i', \pi_{E_{-i}}}$ and the distribution $\rho_{\pi_E}$. Then the trained $D(s, \mathbf{a})$ can be used to construct different intrinsic rewards, which will correspond to different choices of $f$-divergence [13, 11, 9]. One major drawback of this paradigm is that the discriminator depends on the new policy $\pi_i'$ and needs re-training once the policy $\pi_i'$ is updated. Another popular paradigm is to learn an intrinsic reward directly from the target distribution $\rho_{\pi_E}$ like the prediction error. Besides using the prediction error, there are also other choices, including energy-based models (EBM) [19] and expert variance [3]. However, those methods usually require specialized training techniques.

## 3.2 Response Diversity via Convex Hull Enlargement

Take the two-player game for example. In games with more than two players, one can simply denote players other than player $i$ as player $-i$. Thanks to the empirical payoff matrix, another fundamental way to characterize the diversity of a new policy is through the reward outcome from the interaction with many different opponents. Each row in the empirical payoff matrix embeds how the corresponding row policy behaves against different opponent policies. We are therefore justified in using the row vector of the empirical payoff matrix to represent the corresponding row policy.

Formally, assume the row player maintains a pool of policies $\mathfrak{P}_i = \{\pi_i^1, \ldots, \pi_i^M\}$ and the column player maintains a pool of policies $\mathfrak{P}_{-i} = \{\pi_{-i}^1, \ldots, \pi_{-i}^N\}$. The induced $(k, j)$ entry in the empirical payoff matrix $\mathbf{A}_{\mathfrak{P}_i \times \mathfrak{P}_{-i}}$ is given by $\phi_i(\pi_i^k, \pi_{-i}^j)$, where the function $\phi_i$ encapsulates the reward outcome for player $i$ given the joint policy $(\pi_i^k, \pi_{-i}^j)$. Now we can define the diversity measure induced by the reward representations:

$$\mathrm{Div}_{\mathrm{rew}}(\pi_i^{M+1}) = D(\mathbf{a}_{M+1} || \mathbf{A}_{\mathfrak{P}_i \times \mathfrak{P}_{-i}}) \tag{4}$$

$$\mathbf{a}_{M+1}^\top := (\phi_i(\pi_i^{M+1}, \pi_{-i}^j))_{j=1}^N . \tag{5}$$

$D(\mathbf{a}_{M+1} || \mathbf{A}_{\mathfrak{P}_i \times \mathfrak{P}_{-i}})$ essentially measures the diversity of the new vector $\mathbf{a}_{M+1}$ given the presence of row vectors in $\mathbf{A}_{\mathfrak{P}_i \times \mathfrak{P}_{-i}}$.

Inspired by the intuition of the convex hull that indicates the representational capacity of a pool of policies, the diverse new policy should seek to enlarge the convex hull of reward vectors as large as possible. To characterize the contribution of a vector to the enlargement of the convex hull directly, we define the novel diversity measure as a form of Euclidean projection:

$$\mathrm{Div}_{\mathrm{rew}}(\pi_i^{M+1}) = \min_{\substack{\mathbf{1}^\top \boldsymbol{\beta} = 1 \\ \boldsymbol{\beta} \geq 0}} || \mathbf{A}_{\mathfrak{P}_i \times \mathfrak{P}_{-i}}^\top \boldsymbol{\beta} - \mathbf{a}_{M+1} ||_2^2 . \tag{6}$$

Unfortunately, there is no closed-form solution to this optimization problem. To facilitate the optimization, we propose a practical and differential lower bound:

**Theorem 2.** *For a given empirical payoff matrix* $\mathbf{A}$ *and the reward vector* $\mathbf{a}_{M+1}$*, the lower bound of* $\mathrm{Div}_{\mathrm{occ}}$ *is given by:*

$$\mathrm{Div}_{\mathrm{rew}}(\pi_i^{M+1}) \geq \frac{\sigma_{\min}^2(\mathbf{A})(1 - \mathbf{1}^\top (\mathbf{A}^\top)^\dagger \mathbf{a}_{M+1})^2}{M} + ||(\mathbf{I} - \mathbf{A}^\top (\mathbf{A}^\top)^\dagger) \mathbf{a}_{M+1}||^2 , \tag{7}$$

*where* $(\mathbf{A}^\top)^\dagger$ *is the Moore–Penrose pseudoinverse of* $\mathbf{A}^\top$*, and* $\sigma_{\min}(\mathbf{A})$ *is the minimum singular value of* $\mathbf{A}$*.*

*Proof.* See Appendix A.2. □

Let $F(\pi_i^{M+1})$ be the right hand of the inequality. Then $F(\pi_i^{M+1})$ serves as a lower bound of $\mathrm{Div}_{\mathrm{rew}}(\pi_i^{M+1})$.

# 4  A Unified Diverse Objective for Best Response

Equipped with the unified diversity measure, we are ready to propose the diversity-aware response during each iteration of PSRO:

$$\arg\max_{\pi_i'} \mathbb{E}_{s, \mathbf{a} \sim \rho_{\pi_i', \pi_{E_{-i}}}} [r(s, \mathbf{a})] + \lambda_1 \mathrm{Div}_{\mathrm{occ}}(\pi_i') + \lambda_2 \mathrm{Div}_{\mathrm{rew}}(\pi_i') . \tag{8}$$

If both $\lambda_1$ and $\lambda_2$ are 0, then objective is a normal best response.

---

**Algorithm 1** Gradient-based Optimization for Unified Diverse Response

---

1: **Input:** population $\mathfrak{P}_i$ for each $i$, meta-game $\mathbf{A}_{\mathfrak{P}_i \times \mathfrak{P}_{-i}}$, state-action dataset $\{(s, \mathbf{a})\}$, weights $\lambda_1$ and $\lambda_2$

2: $\sigma_i, \sigma_{-i} \leftarrow$ Nash on $\mathbf{A}_{\mathfrak{P}_i \times \mathfrak{P}_{-i}}$

3: $\boldsymbol{\pi_E} \leftarrow$ Aggregate according to $\sigma_i, \sigma_{-i}$

4: $r_i^{int}(s, \mathbf{a}) \leftarrow$ Train fixed reward from distribution $(s, \mathbf{a}) \sim \rho_{\boldsymbol{\pi_E}}$ by EBM or prediction errors.

5: $\theta^* \leftarrow$ Train $\pi_i'(\theta)$ against fixed opponent policies $\pi_{E_{-i}}$ by single-agent RL algorithm with $r_i(s, \mathbf{a}) = r_i^{ext}(s, \mathbf{a}) + \lambda_1 r_i^{int}(s, \mathbf{a})$, $r_i^{ext}$ is the original reward function.

6: $\frac{\partial F}{\partial \mathbf{a}_{M+1}} \leftarrow$ Simulate the reward row vector $\mathbf{a}_{M+1}$ using new $\pi_i'(\theta)$ and compute $\frac{\partial F}{\partial \mathbf{a}_{M+1}}$ analytically.

7: $\hat{\theta} \leftarrow$ Train $\pi_i'(\theta^\star)$ against a new mixture distribution $\sigma_{-i} + \lambda_2 \frac{\partial F}{\partial \mathbf{a}_{M+1}}$ of opponent policies.

8: **Output:** policy $\pi_i'(\hat{\theta})$

---

## 4.1 On the Optimization of Diverse Regularizers

**Discussions on Optimizing Behavioral Diversity.** As discussed in Section 3.1, diversity in the occupancy measure level is fully compatible with the reinforcement learning task since the agent can get an intrinsic reward $r_i^{int}(s, \mathbf{a})$ to indicate the novelty of a state-action pair $(s, \mathbf{a})$. Therefore, to optimize the first two items in the objective, we only need to add the original reward by the $\lambda_1$-weighted intrinsic reward. Another issue we need to address is to sample $(s, \mathbf{a})$ from the distribution $\rho_{\boldsymbol{\pi_E}}$, which has been mentioned in Section 3. Since $\boldsymbol{\pi_E}$ is not a true policy but only a hypothetical policy aggregated according to the mixture $(\sigma_i, \sigma_{-i})$, sampling from $\rho_{\boldsymbol{\pi_E}}$ is equivalent to sampling from $\rho_{\pi_i^k, \pi_i^j}$ with probability $\sigma_i^k \sigma_{-i}^j$.

**Discussions on Optimizing Response Diversity.** Optimizing $\text{Div}_{\text{rew}}$ is not so easy since it involves an inner minimization problem. Fortunately, we have derived a closed-form low-bound $F$, which can serve as a surrogate for the outer maximization.

Assume the policy $\pi_i'$ is parameterized by $\theta$ as $\pi_i'(\theta)$. Then the gradient of $F$ with respect to $\theta$ is given by:

$$\frac{\partial F(\pi_i'(\theta))}{\partial \theta} = \frac{\partial \mathbf{a}_{M+1}}{\partial \theta} \frac{\partial F}{\partial \mathbf{a}_{M+1}} = \left( \frac{\partial \phi_i(\pi_i'(\theta), \pi_{-i}^1)}{\partial \theta}, \ldots, \frac{\partial \phi_i(\pi_i'(\theta), \pi_{-i}^N)}{\partial \theta} \right) \frac{\partial F}{\partial \mathbf{a}_{M+1}} .$$

$\frac{\partial F}{\partial \mathbf{a}_{M+1}}$ controls weights of the policy gradient backpropagated from different opponents policies $\pi_{-i}$. For practical implement, we sample an opponent $j$ according to the absolute values of $\frac{\partial F}{\partial \mathbf{a}_{M+1}}$ and then train $\pi_i'$ against the opponent $\pi_{-i}^j$ using gradient descent or ascent, which depends on the sign of the $j_{th}$ entry of $\frac{\partial F}{\partial \mathbf{a}_{M+1}}$.

**Joint Optimization.** One issue worth our notice is that the update direction of $\text{Div}_{\text{rew}}$ will heavily rely on the initialization of $\pi_i'(\theta)$. A bad initialization of $\theta$ will make the response diversity tell $\pi_i'$ to update toward worse rewards. Therefore, we propose to first optimize the first two items in the objective jointly and then optimize $\pi_i'$ using $\text{Div}_{\text{rew}}$. The final unified diverse response with gradient-based optimization is described in Algorithm 1.

In addition to the gradient-based optimization, we also provide other kinds of optimization oracles suitable for different games. Pseudocodes can be found in Appendix G.

## 4.2 Evaluation Metrics

**Exploitability.** Exploitability [16] measures the distance of a joint policy from the Nash equilibrium. It shows how much each player gains by deviating to their best responses:

$$\text{Expl}(\boldsymbol{\pi}) = \sum_{i=1}^{N} (\max_{\pi_i'} \phi_i(\pi_i', \pi_{-i}) - \phi_i(\pi_i, \pi_{-i})) . \tag{9}$$

Therefore, the smaller exploitability means the joint policy is closer to the Nash equilibrium.

**Population Effectivity.** Note the limitation of exploitability is that it only measures how exploitable a single joint policy is. Therefore, to evaluate the effectiveness of a population, we first need to get an aggregated policy from a population, and we usually use the Nash aggregated policy output by PSRO. Since the Nash is computed over the meta game, which varies with the opponents, the aggregation may be sub-optimal and cannot be used to represent a population. Intuitively, the aggregation weights, and further, the evaluation of a population should not be determined by the population that a specific opponent holds. To address this issue, we propose a *generalized opponent-free* concept of exploitability called Population Effectivity (PE) by looking for the optimal aggregation in the worst cases:

$$\text{PE}(\{\pi_i^k\}_{k=1}^N) = \min_{\pi_{-i}} \max_{\substack{\mathbf{1}^\top \boldsymbol{\alpha}=1 \\ \alpha_i \geq 0}} \sum_{k=1}^N \alpha_k \phi_i(\pi_i^k, \pi_{-i}) \,. \tag{10}$$

PE is again a NE over a two-player zero-sum game, where the player owning the population optimizes towards an optimal aggregation denoted by $\boldsymbol{\alpha}$, while the opponent can search over the entire policy space. Similar ideas of revealing the relationship between the minimax objective and diversity can be also found in [40, 34]. Next, we offer a simple example to further illustrate the limitations of exploitability and superiority of PE.

**Example 1.** *Consider the matrix game Rock-Scissor-Paper, where $\phi_1(\pi_1, \pi_2) = \pi_1^\top \mathbf{A} \pi_2$ and $\phi_2(\pi_2, \pi_1) = \pi_2^\top \mathbf{B} \pi_1$, $\pi_1 \in \mathbb{R}^3$, $\pi_2 \in \mathbb{R}^3$, $\mathbf{A} = \begin{bmatrix} 0 & 1 & -1 \\ -1 & 0 & 1 \\ 1 & -1 & 0 \end{bmatrix}$, $\mathbf{B} = -\mathbf{A}^\top$. Suppose player 1 holds the population $\mathfrak{P}_1 = \{ \begin{bmatrix} 1 \\ 0 \\ 0 \end{bmatrix}, \begin{bmatrix} 0 \\ 1 \\ 0 \end{bmatrix}, \begin{bmatrix} 0 \\ 0 \\ 1 \end{bmatrix} \}$, i.e. $\{Rock, Scissor, Paper\}$ and $\mathfrak{P}_2 = \{ \begin{bmatrix} 1 \\ 0 \\ 0 \end{bmatrix} \}$, i.e. $\{Rock\}$. Then the meta-game $\mathbf{A}_{\mathfrak{P}_1 \times \mathfrak{P}_2} = \begin{bmatrix} 0 \\ -1 \\ 1 \end{bmatrix}$. The nash aggregated joint policy $(\pi_1, \pi_2) = (\begin{bmatrix} 0 \\ 0 \\ 1 \end{bmatrix}, \begin{bmatrix} 1 \\ 0 \\ 0 \end{bmatrix})$. Now we can compute $\text{Expl}((\pi_1, \pi_2))$ as:*

$$\text{Expl}((\pi_1, \pi_2)) = \max_{\pi_1'} \phi_1(\pi_1', \pi_2) - \phi_1(\pi_1, \pi_2) + \max_{\pi_2'} \phi_2(\pi_2', \pi_1) - \phi_2(\pi_2, \pi_1) \tag{11}$$

$$= \max_{\pi_1'} \phi_1(\pi_1', \pi_2) + \max_{\pi_2'} \phi_2(\pi_2', \pi_1) = 2. \tag{12}$$

*Now notice that the contribution of player 1 to the exploitability is $\max_{\pi_2'} \phi_2(\pi_2', \pi_1)$, which equals 1. However, it is not reasonable that player 1 and 2 have the same contribution to the exploitability since player 1 has a perfect diverse policy set. Instead, if we use PE as the metric:*

$$\text{PE}(\mathfrak{P}_1) = 0 \,, \tag{13}$$

*which justifies that player 1 has already found a perfect population.*

In the following theorem, we show that PE is a generalized notion of exploitability under certain conditions and has some desirable properties:

**Theorem 3.** *Population effectivity has the following properties:*

*P1. **Equivalence**: If $N = 1$ and the underlying game $\phi_i(\cdot, \cdot)$ is a symmetric two-player zero-sum game, PE is equivalent to exploitability.*

*P2. **Monotonicity**: If there are two populations $\mathfrak{P}_i, \mathfrak{Q}_i$ and $\mathfrak{P}_i \subseteq \mathfrak{Q}_i$, then $\text{PE}(\mathfrak{P}_i) \leq \text{PE}(\mathfrak{Q}_i)$, while the relationship for exploitability of the Nash aggregated policies of $\mathfrak{P}_i$ and $\mathfrak{Q}_i$ may or may not hold.*

*P3. **Tractability**: If the underlying game $\phi_i(\cdot, \cdot)$ is a matrix game, then computing PE is still solving a matrix game.*

*Proof.* See Appendix A.3. □

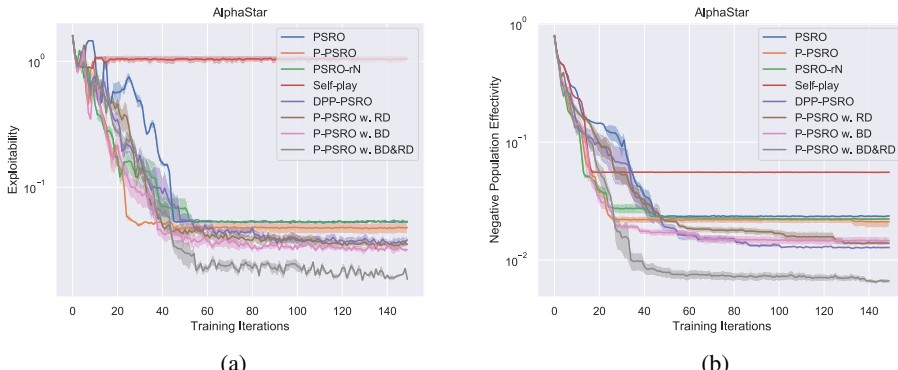

Figure 1: **(a)**:Exploitability *vs.* training iterations (the number of times the optimization oracle is called) on the AlphaStar game. **(b)**: Negative Population Effectivity *vs.* training iterations on the AlphaStar game. Ablation studies of P-PSRO only with BD or RD are also reported.

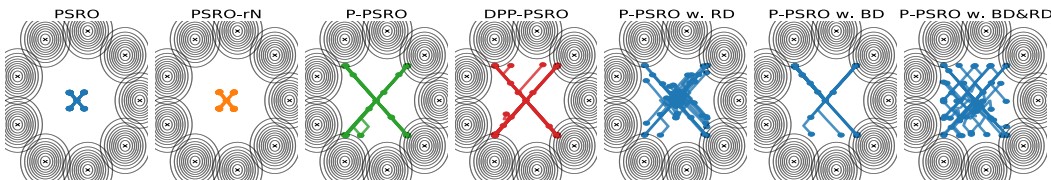

Figure 2: Exploration trajectories during training process on *Non-Transitive Mixture Games*.

## 5 Experiments

To verify that our diversity-regularized best response algorithm can induce a diverse and less exploitable population, we compare our methods with state-of-the-art game solvers including Self-play [12], PSRO [16], PSRO$_{rN}$ [2], Pipeline-PSRO (P-PSRO) [21], DPP-PSRO [24]. In this section, we want to demonstrate the effectiveness of our method to tackle the non-transitivity of zero-sum games, which can be shown via higher PE, lower exploitability, and diverse behaviors. Beyond the simple games, we also have the results on the complex *Google Research Football* game, and our methods can still work. In all the following experiments, we choose the appropriate diversity weights $\lambda_1$ and $\lambda_2$ by extensive hyper-parameter tuning. We also conduct ablation study by choosing different $\lambda_1$ and $\lambda_2$ in Appendix F. The environment details are in Appendix D, and the hyper-parameter settings for each experiment are in Appendix E.3.

**Real-World Games**. [4] studies the properties of some complex real-world games, including AlphaStar and AlphaGO. We test our method on the empirical games generated through the process of solving these real-world games. In Figure 1a, we report the exploitabilities of different algorithms during solving the AlphaStar game, which contains the meta-payoffs for $888$ RL policies. We report values of exploitability and PE during the growth of the population in Figure 1a and Figure 1b. The result shows that with our unified diversity regularizer, our methods achieve the smallest exploitability and largest population effectivity, while most baselines fail to recover the diverse strategies and are easily exploited.

**Non-Transitive Mixture Games**. This zero-sum two-player game consists of $2l+1$ equally-distanced Gaussian humps on the 2D plane. Each player chooses a point in the 2D plane, which will be translated into a $(2l + 1)$-dimensional vector $\pi_i$ with each coordinate being the density in the corresponding Gaussian distribution. The payoff of the game is given by:$\phi_i(\pi_i, \pi_{-i}) = \pi_i^\top \mathbf{S} \pi_{-i} + \mathbf{1}^\top (\pi_i - \pi_{-i})$. According to the delicately designed $\mathbf{S}$, this game involves both the transitive component and non-transitive component, which means the optimal strategy should stay close to the center of the Gaussian and explore all the Gaussian distributions equally.

We firstly visualize the exploration trajectories during different algorithms solving the game in Figure 2. It shows that the best response algorithm regularized by both BD and RD achieves the most diverse trajectories. Although our algorithm finds the most diverse trajectories, such superiority is not revealed by the metric of exploitability in the last row of Table 2. On the other hand, we also report the PE values for the final population generated by different algorithms in Table 2. It can found that our unified diversity regularizer can always help PSRO dominate other baselines in terms of

Table 2: PE$\times 10^2$ for populations generated by different methods when encountering opponents with varying strength on *Non-transitive Mixture Games*. The OS (Opponent Strength) associated with the PE represents the strength of the opponent during the process of using PSRO to solve it. More details can be found in Appendix B. We also report the Exploitability$\times 10^2$ for each population in last row.

| PE(OS) | PSRO | PSRO$_{rN}$ | P-PSRO | DPP-PSRO | P-PSRO w. RD | P-PSRO w. BD | P-PSRO w. BD&RD |
|---|---|---|---|---|---|---|---|
| PE(5) | $-2.11 \pm 0.13$ | $-2.11 \pm 0.14$ | $40.20 \pm 0.09$ | $40.49 \pm 0.07$ | $40.42 \pm 0.08$ | $40.19 \pm 0.10$ | $\mathbf{40.54 \pm 0.12}$ |
| PE(10) | $-13.18 \pm 0.28$ | $-13.18 \pm 0.28$ | $29.14 \pm 0.19$ | $29.45 \pm 0.13$ | $29.55 + 0.13$ | $29.05 + 0.21$ | $\mathbf{29.63 \pm 0.26}$ |
| PE(15) | $-31.17 \pm 0.37$ | $-31.17 \pm 0.37$ | $11.03 \pm 0.26$ | $11.49 \pm 0.21$ | $\mathbf{11.63 \pm 0.15}$ | $10.97 \pm 0.29$ | $11.57 \pm 0.33$ |
| PE(20) | $-49.12 \pm 0.23$ | $-49.12 \pm 0.24$ | $-6.78 \pm 0.14$ | $-6.41 \pm 0.10$ | $-6.52 \pm 0.10$ | $-7.03 \pm 0.21$ | $\mathbf{-6.37 \pm 0.24}$ |
| PE(25) | $-54.59 \pm 0.02$ | $-54.59 \pm 0.01$ | $-12.51 \pm 0.05$ | $-12.28 \pm 0.04$ | $-12.42 \pm 0.03$ | $-12.58 \pm 0.02$ | $\mathbf{-12.18 \pm 0.04}$ |
| Expl | $54.66 \pm 0.06$ | $54.90 \pm 0.10$ | $\mathbf{13.21 \pm 0.29}$ | $13.24 \pm 0.33$ | $13.77 \pm 0.40$ | $41.132 \pm 1.06$ | $13.26 \pm 0.24$ |

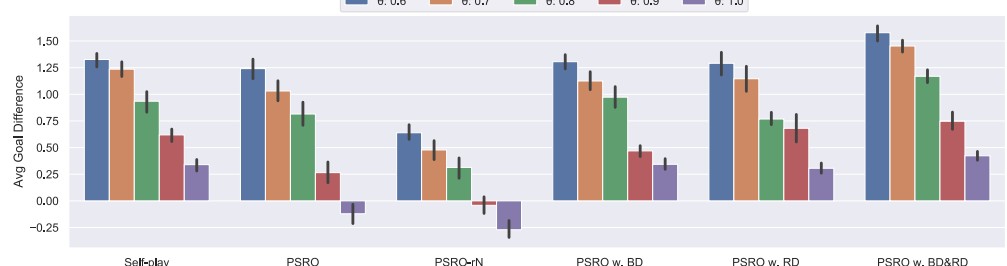

Figure 3: The average goal difference between all the methods and the built-in bots with various difficulty levels $\theta$ ($\theta \in [0, 1]$ and larger $\theta$ means harder bot) on *Google Research Football*.

population effectivity, which also justifies why PE is a better metric to evaluate diverse populations. The details of computing approximate PE using PSRO can be found in Appendix B.

**Google Research Football**. In addition to the experiments on relatively simple games, we also evaluate our methods on a challenging real-world game named *Google Research Football (GRF)* [15]. *GRF* simulates a complete football game under standard rules with 11 players on each team, and a normal match lasts for 3000 steps. The enormous exploration spaces, the long-time horizon, and the sparse rewards problems in this game make it a challenging environment for modern reinforcement learning algorithms. In such complex scenarios, the exploitability of a certain policy or PE of a certain population would be hard to calculate because both metrics involve a max or min operator, and the approximate best response can be quite inaccurate for this complex game. Since our goal is to find robust policies with strong capabilities in real-world games, we compare the average goal differences between the aggregated policies of different methods and the built-in bots with various difficulty levels of *GRF*. The models within each aggregated policies are trained for 300000 steps under the generalized framework of Self-play [12] by selecting opponents according to the probabilities output by different methods.

As depicted in Figure 3, policies trained by PSRO with both BD and RD achieve the largest goal differences when playing against the built-in bots. Moreover, they have an average of **60%** win-rate over other baseline methods (see the table in the Appendix C). We do not report the results of DPP-PSRO since it needs evolutionary updates and cannot scale to such a complex setting. We also abandon the pipeline trick for ease of implement since it does not affect the relative performance among algorithms. Additionally, the discussion of robustness of policies trained with different methods, the network architectures, the hyperparameters, and other detailed experimental setups can also be found in Appendix C.

## 6  Conclusions

This paper investigated a new perspective on unifying diversity measures for open-ended learning in zero-sum games, which shapes an auto-curriculum to induce diverse yet effective behaviors. To this end, we decomposed the similarity measure of MDPs into behavioral and response diversity and showed that most of the existing diversity measures for RL can be concluded into one of the categories of them. We also provided the corresponding diversity-promoting objective and optimization methods, which consist of occupancy measure mismatching and convex hull enlargement. Finally, we proposed population effectivity to overcome the limitation of exploitability in measuring diverse policies for open-ended algorithms. Experimental results demonstrated our method is robust to both highly non-transitive games and complex games like the *Google Research Football* environment.

We also include a discussion on the limitation of our method here. Regarding the limitations from the perspective of the proposed algorithm, One limitation is that the diversity weights $\lambda_1$ and $\lambda_2$ in our paper are manually tuned. Our methods target the non-transitivity in zero-sum games. Therefore, the weights for a game should be related to how strong the non-transitive component is. In future work, we will work towards quantifying the non-transitivity component of a given game automatically and determining the diversity weights correspondingly. Similar ideas have been tested in single-agent RL cases on learning the discount rate [35]. In addition, our methods also inherit the limitations from the framework of PSRO. The advantage of the PSRO based methods also depends on the amount of non-transitivity of the environment [4]. To be specific, if there aren't too many cyclic policies involved in the environment (such as Rock->Paper->Scissor), the newest model generated by the naive self-play training paradigm could probably be the strongest one (a Nash Policy), which implies that a naive self-play would suffice to solve the problem. The PSRO methods will then lose their advantage under such circumstances since they need extra computation resources to maintain meta-payoffs. Regarding the limitations from the perspective of real-world application, the game dynamics could be complex and there could be a lot of randomnesses involved in the real-world games. Thus, the approximated best response trained by reinforcement learning algorithms towards a certain policy could be inaccurate. In the *Google Research Football* experiment, we empirically save the checkpoint when the model win-rate is stable (i.e. the change of win-rate is less than 0.05 during two checks with the check frequency is 1000 model steps) or the training model steps reach an upper bound of 50000. These criteria could be potentially improved or further studied for all the PSRO based methods.

## Acknowledgments

This work is supported by Shanghai Sailing Program (21YF1421900). The authors thank Minghuan Liu for many helpful discussions and suggestions.

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
