# Appendix for "Unifying Behavioral and Response Diversity for Open-ended Learning in Zero-sum Games"

## Table of Contents

## A  Full Proof of Theorems

### A.1  Proof of Theorem 1

To prove Theorem 1, we need the help of the following Lemma

**Lemma 1.** *If $P_{X,Y} = P_X P_{Y|X}$ and $Q_{X,Y} = Q_X P_{Y|X}$ then*

$$D_f(P_{X,Y} || Q_{X,Y}) = D_f(P_X || Q_X). \tag{14}$$

*Proof.* See Proposition 7.1 in [3]. □

Now we can prove our Theorem 1.

*Proof.* For games with only one step (normal-form games, functional-form games), there is only one fixed state. Therefore, the distribution of state-action is equivalent to the distribution of the action. Formally, for $\rho_{\pi'_i, \pi_{E_{-i}}}$, we have:

$$\rho_{\pi'_i, \pi_{E_{-i}}}(s, \mathbf{a}) = (\pi'_i, \pi_{E_{-i}})(s, \mathbf{a}) = \pi'_i(a_i|s)\pi_{-i}(a_{-i}|s) , \tag{15}$$

where the second equation comes from the assumption that policies are independent. Similarly, for $\rho_{\pi_i, \pi_{E_{-i}}}$, we also have:

$$\rho_{\pi_i, \pi_{E_{-i}}}(s, \mathbf{a}) = (\pi_i, \pi_{E_{-i}})(s, \mathbf{a}) = \pi_i(a_i|s)\pi_{-i}(a_{-i}|s) . \tag{16}$$

Therefore, with the help of Lemma 1, we have:

$$D_f(\rho_{\pi'_i, \pi_{E_{-i}}} || \rho_{\pi_i, \pi_{E_{-i}}}) = \mathbb{E}_{s_0 \sim \eta(s)}[D_f(\pi'_i \pi_{-i} || \pi_i \pi_{-i})] = \mathbb{E}_{s_0 \sim \eta(s)}[D_f(\pi'_i(\cdot|s_0) || \pi_i(\cdot|s_0))] \,.$$

$$(17)$$

$\square$

## A.2   Proof of Theorem 2

Let us restate our Theorem 2

**Theorem 2.** *For a given empirical payoff matrix $\mathbf{A} \in \mathbb{R}^{M \times N}$ and the reward vector $\mathbf{a}_{M+1}$ for policy $\pi_i^{M+1}$, the lower bound of $\mathrm{Div}_{\mathrm{occ}}$ is given by:*

$$\mathrm{Div}_{\mathrm{rew}}(\pi_i^{M+1}) \geq \frac{\sigma_{\min}^2(\mathbf{A})(1 - \mathbf{1}^\top (\mathbf{A}^\top)^\dagger \mathbf{a}_{M+1})^2}{M} + ||(\mathbf{I} - \mathbf{A}^\top (\mathbf{A}^\top)^\dagger) \mathbf{a}_{M+1}||^2 \,, \quad (18)$$

*where $(\mathbf{A}^\top)^\dagger$ is the Moore–Penrose pseudoinverse of $\mathbf{A}^\top$, and $\sigma_{\min}(\mathbf{A})$ is the minimum singular value of $\mathbf{A}$.*

*Proof.*

$$\begin{aligned}
\min_{\substack{\mathbf{1}^\top \boldsymbol{\beta}=1 \\ \boldsymbol{\beta} \geq 0}} ||\mathbf{A}^\top \boldsymbol{\beta} - \mathbf{a}_{M+1}||_2^2 &= \min_{\substack{\mathbf{1}^\top \boldsymbol{\beta}=1 \\ \boldsymbol{\beta} \geq 0}} ||\mathbf{A}^\top \boldsymbol{\beta} - \mathbf{A}^\top (\mathbf{A}^\top)^\dagger \mathbf{a}_{M+1}||^2 + ||(\mathbf{I} - \mathbf{A}^\top (\mathbf{A}^\top)^\dagger) \mathbf{a}_{M+1}||^2 \\
&\geq \min_{\mathbf{1}^\top \boldsymbol{\beta}=1} ||\mathbf{A}^\top \boldsymbol{\beta} - \mathbf{A}^\top (\mathbf{A}^\top)^\dagger \mathbf{a}_{M+1}||^2 + ||(\mathbf{I} - \mathbf{A}^\top (\mathbf{A}^\top)^\dagger) \mathbf{a}_{M+1}||^2 \\
&= \min_{\mathbf{1}^\top \boldsymbol{\beta}=1} ||\mathbf{A}^\top (\boldsymbol{\beta} - (\mathbf{A}^\top)^\dagger \mathbf{a}_{M+1})||^2 + ||(\mathbf{I} - \mathbf{A}^\top (\mathbf{A}^\top)^\dagger) \mathbf{a}_{M+1}||^2 \\
&\geq \sigma_{\min}^2(\mathbf{A}) \min_{\mathbf{1}^\top \boldsymbol{\beta}=1} ||\boldsymbol{\beta} - (\mathbf{A}^\top)^\dagger \mathbf{a}_{M+1}||^2 + ||(\mathbf{I} - \mathbf{A}^\top (\mathbf{A}^\top)^\dagger) \mathbf{a}_{M+1}||^2 \\
&= \frac{\sigma_{\min}^2(\mathbf{A})(1 - \mathbf{1}^\top (\mathbf{A}^\top)^\dagger \mathbf{a}_{M+1})^2}{M} + ||(\mathbf{I} - \mathbf{A}^\top (\mathbf{A}^\top)^\dagger) \mathbf{a}_{M+1}||^2 \,,
\end{aligned}$$

where the first equation comes from that we decompose $\mathbf{a}_{M+1}$ into $\mathbf{A}^\top (\mathbf{A}^\top)^\dagger \mathbf{a}_{M+1} + (\mathbf{I} - \mathbf{A}^\top (\mathbf{A}^\top)^\dagger) \mathbf{a}_{M+1}$. The last equation comes from the analytic calculation of $\min_{\mathbf{1}^\top \boldsymbol{\beta}=1} ||\boldsymbol{\beta} - (\mathbf{A}^\top)^\dagger \mathbf{a}_{M+1}||^2$ using Lagrangian. $\square$

## A.3   Proof of Theorem 3

Now let us first restate the propositions.

**Proposition 1.** *If $N = 1$ and the underlying game $\phi_i(\cdot, \cdot)$ is a symmetric two-player zero-sum game, PE is equivalent to exploitability.*

To prove this, let us prove the following Lemma 2.

**Lemma 2.** *For any policy $\pi$ in two-player symmetric zero-sum games:*

$$\phi_i(\pi, \pi) = 0 \,. \quad (19)$$

*Proof.* To begin with, due to the assumption that the game is symmetric, we get:

$$\phi_i(\pi_i, \pi_{-i}) = \phi_{-i}(\pi_{-i}, \pi_i) \,. \quad (20)$$

Since the game is also zero-sum, we have:

$$\phi_{-i}(\pi_{-i}, \pi_i) = -\phi_i(\pi_{-i}, \pi_i) \,. \quad (21)$$

By combing Equation 20 and Equation 33, for any $\pi_i$ and $\pi_{-i}$, we get:

$$\phi_i(\pi_i, \pi_{-i}) + \phi_i(\pi_{-i}, \pi_i) = 0 \,. \quad (22)$$

Let $\pi_i = \pi_{-i} = \pi$, we get what we need to prove:

$$\phi_i(\pi, \pi) = 0 \,. \quad (23)$$

$\square$

Now we can begin proof of our proposition.

*Proof.* To prove this theorem, we need a further assumption that PSRO maintains only one population for two-player symmetric game, which is a quite common practice. Therefore, the joint Nash aggregated policy satisfies that $\boldsymbol{\pi_E} = (\pi_i, \pi_{-i})$ satisfies $\pi_i = \pi_{-i}$. Therefore, with the help of Lemma 2:

$$\phi_i(\pi_i, \pi_{-i}) = \phi_{-i}(\pi_{-i}, \pi_i) = 0 \ . \tag{24}$$

Furthermore, exploitability for symmetric zero-sum game can be written as:

$$\mathrm{Expl}(\boldsymbol{\pi_E}) = \sum_{i=1}^{2} \max_{\pi_i'} \phi_i(\pi_i', \pi_{-i}) - \phi_i(\pi_i, \pi_{-i}) \tag{25}$$

$$= \sum_{i=1}^{2} \max_{\pi_i'} \phi_i(\pi_i', \pi_{-i}) \tag{26}$$

$$= 2 \max_{\pi_i'} \phi_i(\pi_i', \pi_{-i}) \ , \tag{27}$$

where the last equation comes from the symmetry of the game and $\pi_i = \pi_{-i}$.

For PE, it is calculated as:

$$\mathrm{PE}(\{\pi_i\}) = \min_{\pi_{-i}'} \phi_i(\pi_i, \pi_{-i}') \tag{28}$$

$$= -\max_{\pi_{-i}'} \phi_i(\pi_{-i}', \pi_i) \tag{29}$$

$$= -\frac{1}{2} \mathrm{Expl}(\boldsymbol{\pi_E}) \ . \tag{30}$$

The second to last equation comes from Equation 22, and the last equation is due to the assumption that $\pi_i = \pi_{-i}$. $\qquad\square$

**Proposition 2.** *If there are two populations $\mathfrak{P}_i$, $\mathfrak{Q}_i$ and $\mathfrak{P}_i \subseteq \mathcal{Q}_i$, then $\mathrm{PE}(\mathfrak{P}_i) \leq \mathrm{PE}(\mathfrak{Q}_i)$, while the relationship for exploitability of the Nash aggregated policies of $\mathfrak{P}_i$ and $\mathfrak{Q}_i$ may or may not hold.*

*Proof.* We begin with proof of the monotonicity of PE. W.o.l.g, let us assume that $\mathfrak{P}_i = \{\pi_i^k\}_{k=1}^{M}$, $\mathfrak{Q}_i = \{\pi_i^k\}_{k=1}^{N}$, where $M \leq N$. Then for the population effectivity of $\mathfrak{Q}_i$:

$$\mathrm{PE}(\mathfrak{Q}_i) = \min_{\pi_{-i}} \max_{\substack{\mathbf{1}^\top \boldsymbol{\alpha}=1 \\ \alpha_i \geq 0}} \sum_{k=1}^{N} \alpha_k \phi_i(\pi_i^k, \pi_{-i}) \ . \tag{31}$$

where $\boldsymbol{\alpha} = (\alpha_1, \cdots, \alpha_N)^\top$. Let $\alpha_i = 0$ for $M+1 \leq i \leq N$, then we get:

$$\min_{\pi_{-i}} \max_{\substack{\mathbf{1}^\top \boldsymbol{\alpha}=1 \\ \alpha_i \geq 0}} \sum_{k=1}^{N} \alpha_k \phi_i(\pi_i^k, \pi_{-i}) \geq \min_{\pi_{-i}} \max_{\substack{\mathbf{1}^\top \boldsymbol{\alpha}=1, \alpha_i \geq 0 \\ \alpha_i=0 \ \forall M+1 \leq i \leq N}} \sum_{k=1}^{N} \alpha_k \phi_i(\pi_i^k, \pi_{-i}) \tag{32}$$

$$= \min_{\pi_{-i}} \max_{\substack{\mathbf{1}^\top \boldsymbol{\alpha'}=1 \\ \alpha_i' \geq 0}} \sum_{k=1}^{M} \alpha_k' \phi_i(\pi_i^k, \pi_{-i}) \tag{33}$$

$$= \mathrm{PE}(\mathfrak{P}_i) \ . \tag{34}$$

Now we conclude that:

$$\mathrm{PE}(\mathfrak{Q}_i) \geq \mathrm{PE}(\mathfrak{P}_i) \tag{35}$$

Regarding exploitability, the analysis comes from our Example 1. Suppose player 1 holds the population $\mathfrak{P}_1 = \{ \begin{bmatrix} \frac{1}{2} \\ \frac{1}{2} \\ 0 \end{bmatrix} \}$ and $\mathfrak{P}_2 = \mathfrak{P}_1$. Apparently, the Nash aggregated joint policy is

$$\boldsymbol{\pi_E^{\mathfrak{P}}} = ( \begin{bmatrix} \frac{1}{2} \\ \frac{1}{2} \\ 0 \end{bmatrix}, \begin{bmatrix} \frac{1}{2} \\ \frac{1}{2} \\ 0 \end{bmatrix} ) \ , \tag{36}$$

since there is only one policy in each players' population.

Now consider another two populations $\mathfrak{Q}_1 = \{ \begin{bmatrix} \frac{1}{2} \\ \frac{1}{2} \\ 0 \end{bmatrix}, \begin{bmatrix} 0 \\ 1 \\ 0 \end{bmatrix} \}$ and $\mathfrak{Q}_2 = \mathfrak{Q}_1$. Then the Nash aggregated joint policy is given by:

$$\boldsymbol{\pi}_E^{\mathfrak{Q}} = ( \begin{bmatrix} 0 \\ 1 \\ 0 \end{bmatrix}, \begin{bmatrix} 0 \\ 1 \\ 0 \end{bmatrix} ). \tag{37}$$

With simple derivations, the exploitability for $\boldsymbol{\pi}_E^{\mathfrak{P}}$ and $\boldsymbol{\pi}_E^{\mathfrak{Q}}$ is:

$$\mathrm{Expl}(\boldsymbol{\pi}_E^{\mathfrak{P}}) = 1. \tag{38}$$

$$\mathrm{Expl}(\boldsymbol{\pi}_E^{\mathfrak{Q}}) = 2. \tag{39}$$

Now we can conclude that for player 1 and player 2, even their population both get strictly enlarged ($\mathfrak{P}_1 \subseteq \mathfrak{Q}_1$ and $\mathfrak{P}_2 \subseteq \mathfrak{Q}_2$), they become more exploitable: $\mathrm{Expl}(\boldsymbol{\pi}_E^{\mathfrak{Q}}) \geq \mathrm{Expl}(\boldsymbol{\pi}_E^{\mathfrak{P}})$. □

**Proposition 3.** *If the underlying game $\phi_i(\cdot, \cdot)$ is a matrix game, then computing PE is still solving a matrix game.*

*Proof.* The proof follows some simple algebric manipulations. Note that for matrix games, the reward function is given by:

$$\phi_i(\pi_i, \pi_{-i}) = \pi_i^\top \mathbf{P} \pi_{-i} , \tag{40}$$

where $\mathbf{P}$ is the payoff matrix. Then for population effectivity:

$$\mathrm{PE}(\{\pi_i^k\}_{k=1}^N) = \min_{\pi_{-i}} \max_{\substack{\mathbf{1}^\top \boldsymbol{\alpha}=1 \\ \alpha_i \geq 0}} \sum_{k=1}^N \alpha_k \phi_i(\pi_i^k, \pi_{-i}) \tag{41}$$

$$= \min_{\pi_{-i}} \max_{\substack{\mathbf{1}^\top \boldsymbol{\alpha}=1 \\ \alpha_i \geq 0}} \sum_{k=1}^N \alpha_k (\pi_i^k)^\top \mathbf{P} \pi_{-i} \tag{42}$$

$$= \min_{\pi_{-i}} \max_{\substack{\mathbf{1}^\top \boldsymbol{\alpha}=1 \\ \alpha_i \geq 0}} \boldsymbol{\alpha}^\top (\pi_i^1, \cdots, \pi_i^N)^\top \mathbf{P} \pi_{-i} . \tag{43}$$

Therefore, solving PE is still a matrix game with payoff matrix $(\pi_i^1, \cdots, \pi_i^N)^\top \mathbf{P}$. □

## B  PE Approximation via PSRO

We have already mentioned the tractability for PE of matrix games in Theorem 3. However, for more general games, solving exact PE is still very hard. Since PE is still computing an NE, we here propose using PSRO again as the approximate solver. The only difference is that population of player $i$ is already fixed by $\mathfrak{P}_i = \{\pi_i^k\}_{k=1}^N$. Therefore, during iterations of PSRO, only player $-i$ needs to enlarge its population. We now outline the algorithm PE($n$) in Algorithm 2. The intuition behind this algorithm is that the opponent is enlarging its population gradually and trying to exploit $\mathfrak{P}_i$. Therefore, the metric of PE is actually testing how exploitable a population is **by gradually constructing a real adversarial**! The opponent strength $n$ essentially represents how accurate each best response is.

## C  Additional Experimental Results on Google Research Football

In real world games, we expect our models are robust enough to defeat all the previous models in the model pool and show diverse behaviors to better exploit the opponents. To further evaluate the performance of all the models generated with different methods during the training process, we rank all the models with the Elo rating system [2], and the results are shown in Figure 4. It can be found

---

**Algorithm 2** PE($n$)

---

1: **Input:** Population $\mathfrak{P}_i = \{\pi_i^k\}_{k=1}^N$, Opponent Strength $n$, Number of iteraions $T$
2: $\mathfrak{P}_{-i} \leftarrow$ Initialize opponent population with one random policy
3: $\mathbf{A}_{\mathfrak{P}_i \times \mathfrak{P}_{-i}} \leftarrow$ Initialize empirical payoff matrix
4: **for** $t = 1$ **to** $T$ **do**
5:     $\sigma_i, \sigma_{-i} \leftarrow$ Nash Equilibrium on $\mathbf{A}_{\mathfrak{P}_i \times \mathfrak{P}_{-i}}$
6:     $\pi_{-i}^t(\theta) \leftarrow$ Initialize a new opponent policy
7:     $\theta^\star \leftarrow$ Train $\pi_{-i}^t$ against mixture of $\sum_j \sigma_i^j \pi_i^j$ with $n$ gradient steps
8:     $\mathfrak{P}_{-i} \leftarrow \mathfrak{P}_{-i} \cup \{\pi_{-i}^t(\theta^\star)\}$
9:     Compute missing entries in the evaluation matrix $\mathbf{A}_{\mathfrak{P}_i \times \mathfrak{P}_{-i}}$
10: **end for**
11: **Output:** Nash value on $\mathbf{A}_{\mathfrak{P}_i \times \mathfrak{P}_{-i}}$

---

that models generated by PSRO w. BD&RD outperform other methods and reaches an Elo score of around 1300. This implies that a combination of BD and RD will essentially contribute to the generation of diverse opponents during the training, so that the final models will be more robust and less exploitable since they are more likely to be offered strong diverse opponents and have the chance to learn to defeat them. Additionally, we also visualize the policies of our methods when playing against other baseline methods and verify our methods truly generate diverse behaviors (see https://sites.google.com/view/diverse-psro/).

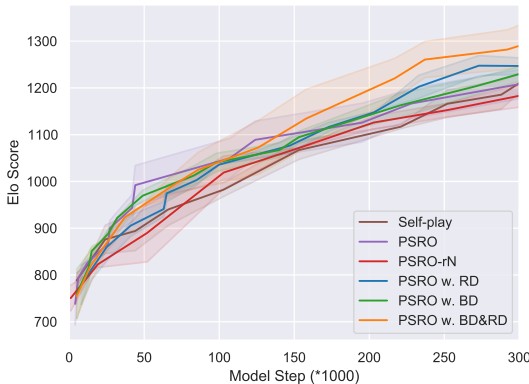

Figure 4: The Elo scores of all the models generated by different methods. Shaded areas represent the standard deviation.

# D  Environment Details

## D.1  Environment Details of Non-Transitive Mixture Model

In our experiments, we set $l = 4$ and use 9 Gaussian distributions in the plane. This environment involves both transitivity and non-transitivity because of the delicately designed $\mathbf{S}$ in the reward function $\phi_i(\pi_i, \pi_{-i}) = \pi_i^\top \mathbf{S} \pi_{-i} + \mathbf{1}^\top(\pi_i - \pi_{-i})$. $\mathbf{S}$ is constructed by:

$$\mathbf{S}[i][k] = \begin{cases} 0 & k = i \\ 1 & 0 < (k-i) \bmod (2l+1) \leq l \\ -1 & \text{otherwise} \end{cases}$$

## D.2  Environment Details of *Google Research Football*

*Google Research Football* (GRF) [4] is a physics-based 3D simulator where agents can be trained to play football. The engine implements a full football game under standard rules (such as goal kicks, side kicks, corner kicks, etc.), with 11 players on each team and 3000 steps duration for a full game. It offers several state wrappers (such as Pixels, SMM, Floats) and the players can be controlled

with 19 discrete actions (such as move in 8 directions, high pass, long pass, steal, etc.). The rewards include both the scoring reward ($+1$ or $-1$) and the checkpoint reward, where the checkpoint reward means that the agent will be reward with $+0.1$ if it is the first time that the agent's team possesses the ball in each of the checkpoint regions.

# E  Hyperparameter Settings

## E.1  Hyperparameter Settings for Real-World Games

We report our hyperparameter setting for real-world metagames in Table 3.

Table 3: The hyperparameters of real-world metagames.

| Parameter | Value | Description |
|---|---|---|
| Learning rate | 0.5 | Learning rate for agents |
| Improvement threshold | 0.03 | Convergence criteria |
| Metasolver | Fictitious play | Method to compute NE |
| Metasolver iterations | 1000 | # of iterations of metasolver |
| Metasolver iterations for PE | 2000 | # of iterations to compute PE |
| # of threads in pipeline | 1.0 | Number of learners in Pipeline-PSRO |
| # of seeds | 5 | # of trials |
| $\lambda_1$ | 0.2 | Weight for BD |
| $\lambda_2$ | 0.2 | Weight for RD |

## E.2  Hyperparameter Settings for Non-Transitive Mixture Model

We report our hyperparameter setting for non-transitive mixture model in Table 4.

Table 4: The hyperparameters of non-transitive mixture model.

| Parameter | Value | Description |
|---|---|---|
| Learning rate | 0.1 | Learning rate for agents |
| Optimizer | Adam | Gradient-based optimization |
| Betas | (0.9, 0.99) | Parameter for Adam |
| $N_{train}$ | 5 | # of iterations using GD per BR |
| $\pi_i$ | $\pi_k^i = \exp(-(x_i - \mu_k)^\top \Sigma (x_i - \mu_k)/2)$ | Policy parameterization |
| $\Sigma$ | $1/2\mathbf{I}$ | Covariance of each Gaussian |
| $u_k$ | $r(\cos(\frac{2\pi}{2l+1}k), \sin(\frac{2\pi}{2l+1}k))$ | Center of each Gaussian |
| $l$ | 4 | 9 Gaussian distributions |
| $r$ | 5 | Radius of each Gaussian |
| Metasolver | Fictitious play | Method to compute NE |
| Metasolver iterations | 1000 | # of iterations of metasolver |
| # of threads in pipeline | 1.0 | Number of learners in Pipeline-PSRO |
| # of iteration | 50 | # of training iterations for PSRO |
| # of seeds | 5 | # of trials |
| $\lambda_1$ | 1 | Weight for BD |
| $\lambda_2$ | 1500 | Weight for RD |
| Decrease rate of $\lambda_1$ and $\lambda_2$ | $1 - \frac{0.7}{1+\exp(-0.25(t-25))}$ | The weights will decrease as the iteration progresses, where $t$ is the current iteration |
| # of iteration for PE | 30 | # of training iterations using PSRO for PE |

## E.3  Hyperparameter Settings for *Google Research Football*

**States and Network Architecture.** For GRF, We use a structured multi-head vector as the states input. The information of each head is listed in Table 5:

The network structure is shown in Figure 5. The shapes of the fully-connected layers for the input heads are: $[32, 64, 64, 16, 16, 128 \times 64, 128 \times 64, 128 \times 128, 128 \times 128, 64, 16, 64, 64]$, followed by three fully-connected layers (i.e. $[512 \times 256 \times 128]$) and finally output the policy and value.

**Hyperparameter Settings for Reinforcement Learning Oracle.** We use IMPALA [1] as the reinforcement learning algorithm to approximate the best response for each opponent selected by different methods during the training process. The hyperparameters are listed in Table 6.

Table 5: The states input for *Google Research Football*.

| Head index | Length | Information |
|---|---|---|
| 0 | 9 | Ball information (position, direction, rotation) |
| 1 | 25 | Ball owner information (ball owned team id, ball owned player id) |
| 2 | 25 | Active player information (id, position, direction, area of the field) |
| 3 | 6 | Active player vs. ball (distance, 1/distance) |
| 4 | 4 | Active player vs. ball player (distance, 1/distance) |
| 5 | 66 | Active player vs. self-team players (position, distance, 1/distance, position cosine, direction cosine) |
| 6 | 66 | Active player vs. oppo-team players (position, distance, 1/distance, position cosine, direction cosine) |
| 7 | 88 | Self-team information (position, direction, tired factor, yellow card, active player, offside flag) |
| 8 | 88 | Oppo-team information (position, direction, tired factor, yellow card, active player, offside flag) |
| 9 | 32 | Goal keeper information (distance to self-player/oppo-player, nearest/farthest player information) |
| 10 | 7 | Game mode information (one-hot) |
| 11 | 29 | Legal action and sticky action information |
| 12 | 76 | History (one-hot) actions of last four steps |

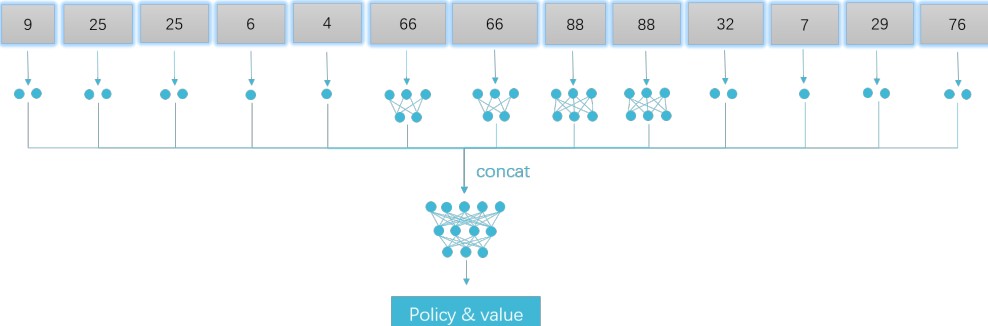

Figure 5: The shape of input states for each head and the general network structure.

**Network Training Details.** We carry out the experiments on six servers (CPU: AMD EPYC 7542 128-Core Processor, RAM: 500G), with each one corresponding to one of six methods (i.e. Self-play, PSRO, PSROrN, PSRO w. BD, PSRO w. RD, PSRO w.BD&RD). For each experiment, the approximated best response (i.e. checkpoint) is saved only when the win-rate against corresponding opponent is stable during two checks (check frequency = 1000 model steps, and $\Delta_{winrate} < 0.05$) or the training model step reaches an upper bound (i.e. 50000 model steps). The $\lambda_1$ and $\lambda_2$ we used for both coefficients are $0.5$. For the *Google Research Football* environment settings, we use both scoring reward and checkpoint reward for the training.

# F   Ablation Studies

We also conduct ablation study on the sensitivity of the diversity weights $\lambda_1$ and $\lambda_2$ in real-world games, non-transitive mixture model, and *Google Research Football*.

## F.1   Real-World Games

We report the exploitability and PE by varying $\lambda_1$ in Figure 6a, 6b and $\lambda_2$ in Figure 7a, 7b. It can be found that too large weights can cause the slow convergence and too small weights prevent the algorithm from finding populations with smaller exploitability and larger PE.

## F.2   Non-Transitive Mixture Model

We report the exploitability of the final population generated by our algorithm with different $\lambda_1$ in Table 7 and $\lambda_2$ in Table 8. In this game, we set both $\lambda_1$ and $\lambda_2$ to decrease following the rate $1 - \frac{0.7}{1+e^{(-0.25(t-25))}}$, where $t$ is the current iteration. We can find that in terms of exploitability, PSRO with only BD cannot help the population to achieve lower exploitability.

Table 6: The hyperparameters of the IMPALA algorithm.

| Parameter | Value |
|---|---|
| Batch Size | 1024 |
| Discount Factor ($\gamma$) | 0.993 |
| Learning Rate | 0.00019896 |
| Number of Actors | 100 |
| Optimizer | Adam |
| Unroll Length/n-step | 1.0 |
| Entropy Coefficient | 0.0001 |
| Value Function Coefficient | 1.0 |
| Grad Clip Norm | 0.5 |
| Rho (for V-Trace) | 1.0 |
| C (for V-Trace) | 1.0 |
| $\lambda_1$ (Weight for BD) | 0.5 |
| $\lambda_2$ (Weight for RD) | 0.5 |

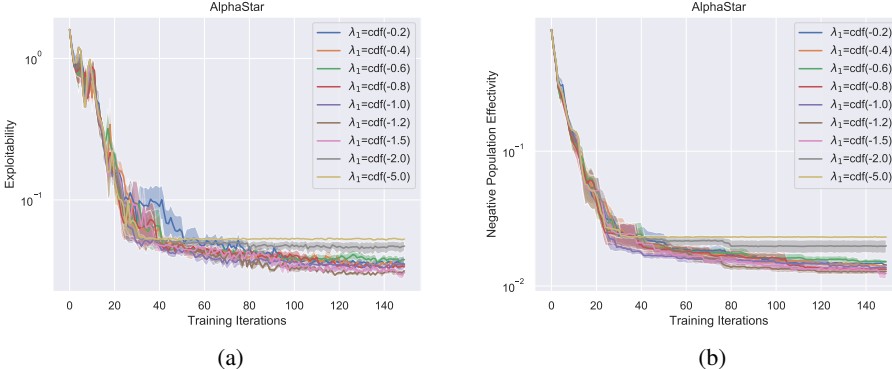

(a)          (b)

Figure 6: Ablation study on $\lambda_1$. **(a)**:Exploitability *vs.* training iterations. **(b)**: Negative Population Effectivity *vs.* training iterations on the AlphaStar game. $\mathrm{cdf}(k) = \int_{-\infty}^{k} \frac{1}{\sqrt{2\pi}} e^{-\frac{x^2}{2}} dx$ is the cumulative distribution function of the standard normal distribution.

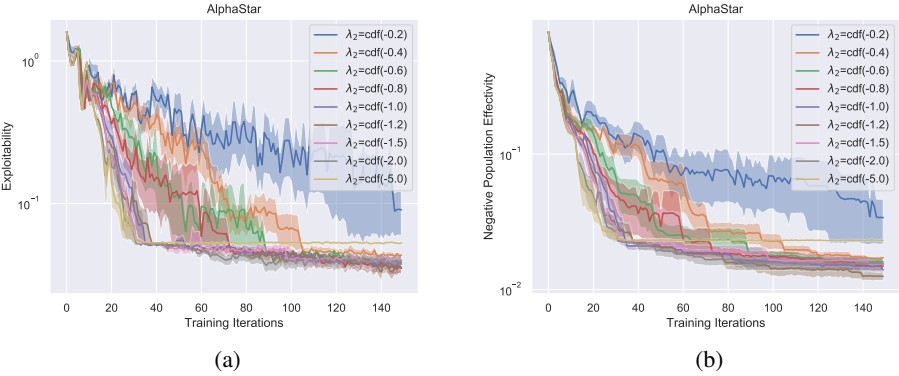

(a)          (b)

Figure 7: Ablation study on $\lambda_2$. **(a)**:Exploitability *vs.* training iterations. **(b)**: Negative Population Effectivity *vs.* training iterations on the AlphaStar game. $\mathrm{cdf}(k) = \int_{-\infty}^{k} \frac{1}{\sqrt{2\pi}} e^{-\frac{x^2}{2}} dx$ is the cumulative distribution function of the standard normal distribution.

Table 7: Exploitability$\times 10^2$ for populations generated by PSRO only with BD with varied diversity weight $\lambda_1$.

| $\lambda_2$ | 7.5 | 15 | 75 | 750 | 1500 | 7500 |
|---|---|---|---|---|---|---|
| Expl | $\mathbf{14.57 \pm 0.69}$ | $14.93 \pm 1.87$ | $14.64 \pm 1.48$ | $42.37 \pm 10.12$ | $33.39 \pm 5.71$ | $62.69 \pm 10.90$ |

Table 8: Exploitability$\times 10^2$ for populations generated by PSRO only with RD with varied diversity weight $\lambda_2$.

| $\lambda_2$ | 0.5 | 1.0 | 5.0 | 10.0 | 50.0 |
|---|---|---|---|---|---|
| Expl | $16.23 \pm 0.48$ | $\mathbf{14.06 \pm 1.20}$ | $14.77 \pm 0.09$ | $15.60 \pm 1.11$ | $31.29 \pm 12.93$ |

Table 9: The win-rate between the final policies of different methods after trained for 300000 model steps. (We set $\lambda_1 = \lambda_2 = 0.5$ as default values for PSRO w. RD and PSRO w. BD&RD)

| Method | Self-play | PSRO | PSRO$_{rN}$ | PSRO w. BD | PSRO w. RD | PSRO w. BD&RD |
|---|---|---|---|---|---|---|
| PSRO w. RD ($\lambda_2 = 1.0$) | $0.62 \pm 0.01$ | $0.49 \pm 0.03$ | $0.65 \pm 0.02$ | $0.47 \pm 0.01$ | $0.28 \pm 0.04$ | $0.33 \pm 0.02$ |
| PSRO w. RD ($\lambda_2 = 0.5$) | $0.68 \pm 0.03$ | $\mathbf{0.61 \pm 0.02}$ | $0.74 \pm 0.03$ | $0.54 \pm 0.02$ | - | $0.43 \pm 0.02$ |
| PSRO w. RD ($\lambda_2 = 0.2$) | $0.63 \pm 0.02$ | $0.48 \pm 0.02$ | $0.68 \pm 0.02$ | $0.50 \pm 0.01$ | $0.45 \pm 0.03$ | $0.28 \pm 0.03$ |
| PSRO w. BD&RD | $\mathbf{0.74 \pm 0.02}$ | $\mathbf{0.78 \pm 0.01}$ | $\mathbf{0.80 \pm 0.05}$ | $\mathbf{0.69 \pm 0.02}$ | $\mathbf{0.57 \pm 0.02}$ | - |

---

**Algorithm 3** Optimization for Matrix Games

---

1: **Input:** population $\mathfrak{P}_i$ for each $i$, meta-game $\mathbf{A}_{\mathfrak{P}_i \times \mathfrak{P}_{-i}}$, weights $\lambda_1$ and $\lambda_2$, learning rate $\mu$
2: $\sigma_i, \sigma_{-i} \leftarrow$ Nash on $\mathbf{A}_{\mathfrak{P}_i \times \mathfrak{P}_{-i}}$
3: $\boldsymbol{\pi_E} \leftarrow$ Aggregate according to $\sigma_i, \sigma_{-i}$
4: $\pi'_i(\theta) \leftarrow$ Initialize a new random policy for player $i$
5: $\mathbf{BR}_{qual} \leftarrow$ Compute best response against mixture of opponents $\sum_k \sigma^k_{-i} \phi_i(\cdot, \pi^k_{-i})$
6: **while** the reward $p$ improvement does not meet the threshold **do**
7:   $\mathbf{BR}_{occ} \leftarrow \arg\max_{s_j} D_f(s_j \| \pi_i)$ for each pure strategy $s_j$
8:   $\mathbf{BR} \leftarrow$ Choose $\mathbf{BR} = \mathbf{BR}_{occ}$ with probability $\lambda_1$ else $\mathbf{BR} = \mathbf{BR}_{qual}$
9:   $\theta \leftarrow \mu\theta + (1 - \mu)\theta_{\mathbf{BR}}$
10:   $p \leftarrow$ Compute the payoff $p$ after the update according to $\sum_k \sigma^k_{-i} \phi_i(\pi'_i(\theta), \pi^k_{-i})$
11: **end while**
12: $\mathbf{BR}_{rew} \leftarrow \arg\max_{s_j} F(s_j)$ for each pure strategy $s_j$ with probability $\lambda_2$ else $\mathbf{BR}_{qual}$
13: $\theta \leftarrow \mu\theta + (1 - \mu)\theta_{\mathbf{BR}_{rew}}$
14: **Output:** policy $\pi'_i(\hat{\theta})$

---

### F.3 Google Research Football

We also conduct an ablation study on the weight of RD $\lambda_2$ (see Table 9) in the GRF environment, where we fixed $\lambda_1$ to be 0.5 and show the results with different $\lambda_2$.

## G Simplified Optimization Method for Unified Diverse Response

In Algorithm 1, we have outlined using RL as the optimization oracle for approximate best response. However, computing best response in real-world metagames (matrix games) or non-transitive mixture model (differential games) can be simplified, since the $f$-divergence objective can be simplified according to Theorem 1 or the reward function $\phi_i$ is analytically accessible. Now we provide the simplified optimization methods separately for matrix games in Algorithm 3 and differential games in Algorithm 4.

---

**Algorithm 4** Optimization for Differential Games

---

1: **Input:** population $\mathfrak{P}_i$ for each $i$, meta-game $\mathbf{A}_{\mathfrak{P}_i \times \mathfrak{P}_{-i}}$, weights $\lambda_1$ and $\lambda_2$, number of gradient updates $N_{train}$
2: $\sigma_i, \sigma_{-i} \leftarrow$ Nash on $\mathbf{A}_{\mathfrak{P}_i \times \mathfrak{P}_{-i}}$
3: $\boldsymbol{\pi_E} = (\pi_i, \pi_{E_{-i}}) \leftarrow$ Aggregate according to $\sigma_i, \sigma_{-i}$
4: $\pi'_i(\theta) \leftarrow$ Initialize a new random policy for player $i$
5: **for** $j = 1$ **to** $N_{train}$ **do**
6:    $p_j \leftarrow$ Compute payoff against the mixture of opponents $p_j = \sum_k \sigma^k_{-i} \phi_i(\pi'_i, \pi^k_{-i})$
7:    $d^{\text{occ}}_j \leftarrow$ Compute BD $d^{\text{occ}}_j = D_f(\pi'_i || \pi_i)$ as the $f$-divergence between $\pi_i$ and $\pi'_i$
8:    $\mathbf{a}_j \leftarrow$ Compute new reward vector as $\mathbf{a}_j = (\phi_i(\pi'_i, \pi^k_{-i}))^{|\mathfrak{P}_{-i}|}_{k=1}$
9:    $d^{\text{rew}}_j \leftarrow$ Compute the lower bound of RD as $F(\mathbf{a}_j)$ according to Theorem 2
10:   $l_j \leftarrow -(p_j + \lambda_1 d^{\text{occ}}_j + \lambda_2 d^{\text{rew}}_j)$
11:   Update $\theta$ to minimize $l_j$ by backpropagation
12: **end for**
13: **Output:** policy $\pi'_i(\theta)$

---