# OpenReview forum: "Towards Unifying Behavioral and Response Diversity for Open-ended Learning in Zero-sum Games"
_NeurIPS.cc/2021/Conference — NeurIPS 2021 Poster_

### Official Review · Reviewer_1idw · 2021-07-14

**Rating:** 7
**Confidence:** 2

**Summary:**

This paper is concerned with learning in zero-sum Markov games, in which there are often non-transitive dynamics between policies/strategies (e.g., Rock beats Scissors beats Paper beats Rock ...). A common approach to avoid getting stuck in cycles or local optima when training in such domains is to encourage algorithms to discover *diverse* solutions or policies.

This paper discusses and formalises how diversity measures can be split up in two broad categories: **behavioral diversity** measures, which measure whether different policies visit different state-action distributions, and **response diversity** measures, which measure whether different policies obtain different vectors of empirical outcomes against a population of opponents. For each of these two categories, the paper proposes a concrete measure and (approximate) methods to optimise them within a Policy Space Response Oracle (PSRO) framework. Additionally, the paper proposes a new Population Effectivity metrid, which is argued to be more fair and informative for evaluating the effectiveness of a population of policies than the Exploitability metric.

The additional diversity objectives are shown to improve training in several different domains.

**Limitations And Societal Impact:**

Limitations are adequately addressed (for example when approximations to optimisation are necessary).

**Main Review:**

**Originality**: to the best of my knowledge, there is a good amount of novel work in this paper, and an adequate discussion of related work.

**Technical quality**: The technical quality of the paper looks solid to me. I will note here that the most theoretical / math-heavy parts, like the proofs, seem beyond my ability and I would probably not easily spot any mistakes in them if there were any. I do feel that I was able to follow along well with the majority of the paper, at the very least on an intuitive level, and from that point of view I do not see many major issues. I only have the following two remarks:

1) Many parts of the paper refer to **unifying** measures of diversity, or providing a **unified** diversity measure: the title of the paper, the abstract, the introduction, Section 4, the conclusion, etc. Maybe I'm mistaken or forgetting about something here, but my understanding of the word "unifying" seems to be different from what's going on in this paper. When I see "unifying", I think of something more *general* (an algorithm, a model, a theory, etc.) that encapsulates multiple other more specific things (which were often previously viewed as being distinct things) as special cases. For example, I would think of TD(lambda) as an algorithm that unifies several other simple reinforcement learning algorithms, such as Monte-Carlo returns (special case with lambda=1) and one-step returns (special case with lambda=0). I do not see the behavioral and response diversity measures being unified in this paper, since they are essentially still two completely distinct objectives that can simply be added together (as in Equation 8).

2) In tables of results, such as Table 2, when some results are presented in boldface this is often interpreted as those results being the best of their row with statistical significance. But in Table 2, it appears that all the best results of every row are presented in boldface, even if their confidence intervals (or are they standard errors?) are overlapping. For example, in the first row, the 40.49 +- 0.07 interval for DPP-PRSO overlaps with the 40.54 +- 0.12 interval for P-PRSO w. BD&RD, but only the latter is in boldface. In these cases, either no result should be in boldface, or all the results for which the intervals overlap with that of the best should be in boldface.

**Clarity**: the paper is clear, well-written, easy to follow. I will just list a few small typos / points of confusion I found:

- Line 24: Zero-sums --> Zero-sum
- Line 45: why is Maze capitalised? Is this just about any arbitrary maze, or a specific environment named "Maze"?
- Line 255: "and B_i subset of Q_i" --> the Q symbol used here looks different from all the other Qs, is that intentional?
- Page numbers of reference [10] should be 586--593 instead of 1--8
- First point of the Checklist says "See Section ??"

**Significance**: I am not sure if many of the results for the matrix games in Table 2 can be considered significant, but the Google Research Football results in Figure 3 look more convincing. The theoretical insights in terms of diversity measures and Population Effectivity evaluation metric also look promising.

**Time Spent Reviewing:**

2

---

> ### Author Response · Authors · 2021-08-09
> **Response to Reviewer 1idw**
>
> ### We thank the Reviewer 1idw for the two hours of effort and the associated constructive comments that will surely help improve our paper.
>
> 1. > **Reviewer**: Many parts of the paper refer to unifying measures of diversity, or providing a unified diversity measure, but my understanding of the word "unifying" seems to be different from what's going on in this paper.
>
> * **Response**: Firstly, we admit that our methods do not completely unify BD and RD in one principled and fundamental objective, and there might be potential overclaiming by using "unify". Therefore, to correct this, we decide to modify the title as "Towards Unifying ..." to highlight our distributions in laying the ground work for the new objective to be discovered.
>
>     Secondly, we provide some theoretical intuitions about the equivalence and difference between BD and RD. BD is defined as the occupancy measure discrepancy and RD is about diversity in the long-term expected returns. Therefore, we try to build relationships between the difference of two occupancy measures and the difference in the corresponding long-term expected return. Considering two policies $\pi_1$, $\pi_2$ and the associate occupancy measures $\rho_{\pi_1}$, $\rho_{\pi_2}$. We quantify the occupancy measure discrepancy using the integral probability metric (IPM) [4]:
>     $$
>     d_{\mathcal{F}}(\rho_{\pi_1}, \rho_{\pi_2})=\sup_{f\in\mathcal{F}}|E_{(s,a)\sim \rho_{\pi_1}}[f(s, a)]-E_{(s,a)\sim \rho_{\pi_2}}[f(s, a)]|
>     $$
>     If we regard $f(s,a)$ as a kind of reward function of the underlying MDP, [1] tells us that:
>     $$
>     E_{(s,a)\sim \rho_\pi}[f(s, a)]=\sum_{s,a}\rho_\pi(s,a)f(s, a)=\eta_f(\pi),
>     $$
>     where $\eta_f(\pi)$ is the **time-average** expected return of $\pi$ under the reward function $f$. Then we can conclude:
>     $$
>     d_{\mathcal{F}}(\rho_{\pi_1}, \rho_{\pi_2})=\sup_{f\in\mathcal{F}}|\eta_f(\pi_1)-\eta_f(\pi_2)|,
>     $$
>     where the left-hand side difference in occupancy measures is associated with the right-hand side difference in expected return. However, it is hard to reach the $\sup$ exactly. An alternate approximation is:
>     $$
>     d_{\mathcal{F}}(\rho_{\pi_1}, \rho_{\pi_2})\approx\max_{f\in\{f_1, \cdots,f_n\}}|\eta_f(\pi_1)-\eta_f(\pi_2)|,
>     $$
>     Therefore, if we have a diverse reward function set $\{f_1, \cdots,f_n\}$, the approximation can be very accurate. Note that this analysis is based on the single-agent setting.
>     Returning to our multi-agent problem, a diverse reward function set can be achieved by a diverse opponent set since the marginal reward function for the player depends on the opponent policy as:
>     $$
>     f_i(\mathbf{s}, a_i)=\sum_{a_{-i}}r_i(\mathbf{s}, a_{i}, a_{-i})\pi_{-i}(a_{-i}|\mathbf{s}),
>     $$
>     where the LHS is the marginalized reward function of player $i$ for the fixed opponent $\pi_{-i}$. Based on this, we argue that during iterations of our methods, the population becomes more diverse and the gap between the difference in occupancy measures and the difference in the expected return gets closer. Therefore, the effects of BD and RD will become increasingly more equivalent during iterations.
>
> 2. > **Reviewer**: In tables of results, such as Table 2, when some results are presented in boldface this is often interpreted as those results being the best of their row with statistical significance. But in Table 2, it appears that all the best results of every row are presented in boldface, even if their confidence intervals (or are they standard errors?) are overlapping.
>
> * **Response**: We admit the performance of different algorithms may overlap in this game. We will accept your advice of highlighting all the prominent data.
>
> Finally, we appreciate your other careful and helpful suggestions (e.g., typos and writings) and will further revise our manuscript based on them.
>
> [1] Sutton, R. S., McAllester, D. A., Singh, S. P., & Mansour, Y. (2000). Policy gradient methods for reinforcement learning with function approximation. In Advances in neural information processing systems (pp. 1057-1063).

---

### Official Review · Reviewer_QZCS · 2021-07-15

**Rating:** 7
**Confidence:** 4

**Summary:**

This paper focuses on approaches for producing diverse policies in single and multi agent reinforcement learning (RL). The authors first introduce Behavioral and Response diversity, with a single metric to represent each. They then present two mechanisms to evaluate the effectiveness of diverse populations of agents. Finally, they introduce a new algorithm and test it on a series of benchmark environments, showing the performance in terms of the newly introduced objectives as well as overall performance. The paper covers a large amount of material, and overall does a good job in explaining how each component relates, while the results are reasonably strong. My score is only a weak accept, because I believe in the current form it is a useful contribution to the field, but it could be made significantly stronger. There are more details below, but at a high level, it seems there is not actually much unification happening here, and at times the authors seem to re-introduce things. It is worth saying, my score is *more likely to increase than decrease* at this point, but of course that can change given author responses.

_______

Score raised to an accept from a weak accept, on the premise that many of the additional clarifications from the author response are included in the paper.

**Limitations And Societal Impact:**

I could not find a detailed discussion of limitations.

**Main Review:**

Strengths

* The paper addresses an interesting problem, clearly motivated by non-transitive games which have been in focus with agents such as AlphaStar.

* The paper is thorough and rigorous in its definitions, which is clear to follow given sufficient time. The authors introduce their own notation to think about several well known methods and once the reader gets up to speed with this the rest of the paper is clear and interesting.

* The empirical results are clear and seem convincing, with the new approach clearly outperforming a strong set of baselines.

* The definitions of BD and RD make sense to me. This is an intuitive way to think about diversity metrics.

* The hyperparameter ablations are a great addition, compared to many works which do not show this level of granularity.

Questions & Areas for Improvement:

1. The paper fails to deliver on its core promise, to quote the abstract: "work towards offering a unified measure of diversity". Indeed, after categorizing diversity metrics into two classes, I expected to see some analysis explaining when they differ and when they are equivalent. I then expected to see a new metric or objective that provides many or all of the desirable properties of both classes of methods. Instead, the final outcome is rather unsatisfactory - the authors simply add both to the objective. In a sense, this is not unified, this is just adding two (very much distinct) pieces together. I think there are two ways to improve here. The first is to discover the novel diversity objective that truly unifies the methods. However, this is of course very hard. The second, would be to change the title to something like "Towards Unifying... " or "Demystifying..." because those are more appropriate for the contribution of the paper, which in a sense, lays the ground work for the new objective to be discovered. In this second case, I would still hope to see some analysis, either empirical or theoretical, of when these objective succeed and fail.

2. Can you possibly explain the behavior in Figure 2? It seems that DPP-PSRO which uses Response Diversity, has a very similar profile to P-PSRO with Behavioral Diversity, and looks very different to P-PSRO with Response Diversity. Is DPP-PSRO classified correctly in Table 1?

3. In some cases the authors make things a little convoluted, and could explain the theorems in words, as they are often very straightforward observations (e.g. Theorem 1).

4. The population effectivity metric looks like a robustness objective, but this is not mentioned. For example [1] focuses on minimax objectives w.r.t reward and of course this is very related, it is just about generalizing with a different degree of freedom. There is nothing wrong with PE not being 100% novel, it just should be more clearly compared to existing works.

5. Limitations of the method do not seem to be thoroughly discussed anywhere. The checklist says yes, but I couldn't find it.

Minor Comments/Typos:

-l.24: "Zero-sums" -> "zero-sum"

-l.83: "is a principle way" -> "is a principled way"

-l.119: "bisimulation metric" -> "bisimulation metrics"

-l.121: "build the metric" -> "build a metric"

-l.130: "state-action pair" -> "state-action pairs"

-l.166: "energy-based model" -> "energy-based models"

-l.320: "showed the most" -> "showed that most"

References

[1] Zahavy et al. Discovering a set of policies for the worst case reward. ICLR 2021

**Time Spent Reviewing:**

6

---

> ### Author Response · Authors · 2021-08-09
> **Response to Reviewer QZCS**
>
> ### We thank the Reviewer QZCS for the six hours of effort and the associated constructive comments that will surely help improve our paper.
>
> 1. > **Reviewer**: The paper fails to deliver on its core promise, to quote the abstract: "work towards offering a unified measure of diversity".
>
>
> * **Response**: Firstly, we admit that our methods do not completely unify BD and RD in one principled and fundamental objective, and there might be potential overclaiming by using "unify". Therefore, to correct this, we adopt your suggestion to modify the title as "Towards Unifying ..." to highlight our distributions in laying the ground work for the new objective to be discovered.
>
>     Secondly, we provide some theoretical intuitions about the equivalence and difference between BD and RD. BD is defined as the occupancy measure discrepancy and RD is about diversity in the long-term expected returns. Therefore, we try to build relationships between the difference of two occupancy measures and the difference in the corresponding long-term expected return. Considering two policies $\pi_1$, $\pi_2$ and the associate occupancy measures $\rho_{\pi_1}$, $\rho_{\pi_2}$. We quantify the occupancy measure discrepancy using the integral probability metric (IPM) [4]:
>     $$
>     d_{\mathcal{F}}(\rho_{\pi_1}, \rho_{\pi_2})=\sup_{f\in\mathcal{F}}|E_{(s,a)\sim \rho_{\pi_1}}[f(s, a)]-E_{(s,a)\sim \rho_{\pi_2}}[f(s, a)]|
>     $$
>
>     If we regard $f(s,a)$ as a kind of reward function of the underlying MDP, [5] tells us that:
>     $$
>     E_{(s,a)\sim \rho_\pi}[f(s, a)]=\sum_{s,a}\rho_\pi(s,a)f(s, a)=\eta_f(\pi),
>     $$
>     where $\eta_f(\pi)$ is the **time-average** expected return of $\pi$ under the reward function $f$. Then we can conclude:
>     $$
>     d_{\mathcal{F}}(\rho_{\pi_1}, \rho_{\pi_2})=\sup_{f\in\mathcal{F}}|\eta_f(\pi_1)-\eta_f(\pi_2)|,
>     $$
>     where the left-hand side difference in occupancy measures is associated with the right-hand side difference in expected return. However, it is hard to reach the $\sup$ exactly. An alternate approximation is:
>     $$
>     d_{\mathcal{F}}(\rho_{\pi_1}, \rho_{\pi_2})\approx\max_{f\in\{f_1, \cdots,f_n\}}|\eta_f(\pi_1)-\eta_f(\pi_2)|,
>     $$
>     Therefore, if we have a diverse reward function set $\{f_1, \cdots,f_n\}$, the approximation can be very accurate. Note that this analysis is based on the single-agent setting.
>     Returning to our multi-agent problem, a diverse reward function set can be achieved by a diverse opponent set since the marginal reward function for the player depends on the opponent policy as:
>     $$
>     f_i(\mathbf{s}, a_i)=\sum_{a_{-i}}r_i(\mathbf{s}, a_{i}, a_{-i})\pi_{-i}(a_{-i}|\mathbf{s}),
>     $$
>     where the LHS is the marginalized reward function of player $i$ with the fixed opponent $\pi_{-i}$. Based on this, we argue that during iterations of our methods, the population becomes more diverse and the gap between the difference in occupancy measures and the difference in the expected return gets closer. Therefore, the effects of BD and RD will become increasingly more equivalent during iterations.
> 2. > **Reviewer**: It seems that DPP-PSRO which uses Response Diversity, has a very similar profile to P-PSRO with Behavioral Diversity, and looks very different to P-PSRO with Response Diversity. Is DPP-PSRO classified correctly in Table 1?
>
> * **Response**: Yes, DPP-PSRO is classified correctly in Table 1. The core idea of DPP-PSRO is to construct the Determinantal Point Process (DPP) through the empirical payoff vectors, and their regularized objective encourages the new policy towards increasing the expected cardinality of DPP, which lies in the domain of Response Diversity. We hypothesize that the reason why DPP-PSRO looks different from P-PSRO with RD is that the DPP objective has much fewer exploratory effects than our RD objective since the DPP objective does not have a straight relationship with the empirical gamescape and has a large overlap with the ordinary best response objective in this game. Regarding P-PSRO with BD, the BD objective will simply push the new policy to be as far as possible from a specific fixed point (the Nash aggregated policy) in this relatively simple game. Thus, the BD objective built on the occupancy measure discrepancy is less informative in this relatively simple game since there is no complex interaction between the policy and the environment dynamics, which is in the definition of occupancy measure. Therefore, it does not bring up many exploratory effects under this setting either. That is why DPP-PSRO and P-PSRO with BD look similar, and both look like ordinary P-PSRO, considering the fact that we use the approximate best response during iterations via gradient descent, which also brings up a few exploratory behaviors for ordinary P-PSRO in Figure 2.
>
> 3. > **Reviewer**: In some cases the authors make things a little convoluted, and could explain the theorems in words, as they are often very straightforward observations (e.g. Theorem 1).
>
> * **Response**: We apologize for the potential lack of some straightforward and intuitive explanations for some statements. The intuition behind Theorem 1 is that since the game is one-step, the policy and the transition dynamics can be easily decoupled. Therefore, the divergence between occupancy measures can be simplified as the divergence between policies given that the transition dynamics are the same for two policies, We will try to add more intuitions for other propositions and further revise our manuscript.
>
> 4. > **Reviewer**: The population effectivity metric looks like a robustness objective, but [1] is not mentioned.
>
> * **Response**: We highly appreciate the missed reference you mentioned and will include it in our manuscript. We admit that the minimax objective is a common formulation of robustness, and we are actually inspired by this. We also understand there has been extensive literature revealing the relationship between diversity and robustness. For example, [6] shows the equivalence between solving the minimax and the diversity via regularization. Now we discuss the relationship between our metric with the specific objective in [1]. For the minimax objective in [1], the inner minimum is taken over different environment rewards, thus seeking the performance guarantee under the worst environment. For our objective, the minimum is taken over the opponent $\pi_{-i}$, thus seeking the performance guarantee under the strongest opponent, which generalizes with a different degree of freedom compared with [1].
>
> 5. > **Reviewer**: Limitations of the method do not seem to be thoroughly discussed anywhere. The checklist says yes, but I couldn't find it.
>
> * **Response**: We apologize for insufficient discussions on the limitations, and we include more discussions here.
>   * Regarding the limitations from the perspective of the proposed algorithm, One limitation is that the diversity weights $\lambda_1$ and $\lambda_2$ in our paper are manually tuned. Our methods target the non-transitivity in zero-sum games. Therefore, the weights for a game should be related to how strong the non-transitive component is. In future work, we will work towards quantifying the non-transitivity component of a given game automatically and determining the diversity weights correspondingly. Similar ideas have been tested in single-agent RL cases on learning the discount rate [3]. In addition, our methods also inherit the limitations from the framework of PSRO. The advantage of the PSRO based methods also depends on the amount of non-transitivity of the environment [1]. To be specific, if there aren’t too many cyclic policies involved in the environment (such as Rock->Paper->Scissor), the newest model generated by the naive self-play training paradigm could probably be the strongest one (a Nash Policy), which implies that a naive self-play would suffice to solve the problem. The PSRO methods will then lose their advantage under such circumstances since they need extra computation resources to maintain meta-payoffs.
>   * Regarding the limitations from the perspective of real-world application, the game dynamics could be complex and there could be a lot of randomnesses involved in the real-world games. Thus, the approximated best response trained by reinforcement learning algorithms towards a certain policy could be inaccurate. In the Google Research Football experiment, we empirically save the checkpoint when the model win-rate is stable (i.e. the change of win-rate is less than 0.05 during two checks with the check frequency is 1000 model steps) or the training model steps reach an upper bound of 50000. These criteria could be potentially improved or further studied for all the PSRO based methods.
>
> We appreciate your other careful and helpful suggestions (e.g., typos and writings) and will further revise our manuscript based on them.
>
> [1] Zahavy et al. Discovering a set of policies for the worst case reward. ICLR 2021
>
> [2] Czarnecki, Wojciech Marian, et al. "Real world games look like spinning tops." arXiv preprint arXiv:2004.09468 (2020).
>
> [3] Xu, Zhongwen, Hado van Hasselt, and David Silver. “Meta-gradient reinforcement learning.” arXiv preprint arXiv:1805.09801 (2018).
>
> [4] Alfred Müller. Integral probability metrics and their generating classes of functions. Advances in Applied Probability, 29(2):429–443, 1997.
>
> [5] Sutton, R. S., McAllester, D. A., Singh, S. P., & Mansour, Y. (2000). Policy gradient methods for reinforcement learning with function approximation. In Advances in neural information processing systems (pp. 1057-1063).
>
> [6] Xu, H., & Mannor, S. (2012). Robustness and generalization. Machine learning, 86(3), 391-423.

---

> > ### Comment · Reviewer_QZCS · 2021-08-11
> > **Thank you for clarifying**
> >
> > I have read through the comments and overall am satisfied with the responses. Below are some specifics but overall, I hope it is possible to include some more of this discussion in the main body fo the paper.
> >
> > **1)**. "Towards Unifying" is definitely better, while the statements you provide make sense and it would be great if you can include them in Section 3 somehow.
> >
> > **2)**. This makes some sense, it would be great to include.
> >
> > **3)**. This wasn't a massive issue just a note.
> >
> > **4)**. I don't quite buy that the objective is different to [1]. In the answer to 1 you said:
> >
> > *"diverse reward function set can be achieved by a diverse opponent set"*
> >
> > then in 4 you said:
> >
> > *"For the minimax objective in [1], the inner minimum is taken over different environment rewards, thus seeking the performance guarantee under the worst environment. For our objective, the minimum is taken over the opponent"*
> >
> > so, in one case it is fine to say the environment reward and opponent are equivalent, whereas in the other you say they are different? I think it would be useful to be consistent with this, in which case [1] is very related and maybe not a "different degree of freedom". The reduction in novelty from having a similar objective is not important in my opinion.
> >
> > **5)**. All these limitations are great. Please do include them in the paper.
> >
> > I realize it will be a challenge to include all of this with the page limits, but it would be fantastic if as much as possible can be included. As I said before, the score was more likely to increase, so now these have been clarified I am **raising to an accept**. Great work!

---

### Official Review · Reviewer_Rpgw · 2021-07-16

**Rating:** 8
**Confidence:** 3

**Summary:**

This work proposes a novel framework for measuring diversity in (open-ended) multi-agent learning. Using this framework, they formulate objectives to encourage diversity at the policy and policy population level, achieving low exploitability, high population effectivity, and overall good performance in simple as well as relatively complex MARL problems.

**Limitations And Societal Impact:**

Strictly speaking, I don't think the manuscript provides a good overview of the limitations of the proposed framework, nor of any particular weakness in the experimental setting used.

Analysis on societal impact is also very limited (if not non-existent).

I would personally enjoy a discussion at the very least on what other types of losses could be potentially steam from utilising the BD / RD decompositions.

**Main Review:**

This is great work!

### Clarity

The paper reads beautifully. I wasn't very familiar with the open-ended learning take on MARL, but the manuscript provided both a good and clear introduction to the topic as well as a very clean (and seemingly novel!) view on the problem. I thoroughly enjoyed going through it.

### Originality & Significance

I'm not extremely familiar with most on the existing literature on estimating diversity in MARL settings, but the overall proposed analysis framework as well as the obtained losses seem novel and well sound. *At the very least* the manuscript seems significant in bringing a fresh perspective on a few recent methods for population-driven MARL that have helped with generating very successful RL stories.

### Questions

L154:158: How does this actually differ form RED (or RND even) in practice? Furthermore, once the network is learnt on the dataset, does it remain fixed throughout the duration of the experiment, or does it get periodically re-learnt?

L169:170: Shouldn't $-i$ be assignable to a single opponent player? What value does it take in say a game with 3 or 4 players?

### Typos

L45: "like, in Maze" -- looks like a bad copy-paste.

L169: s/for an example/for example/

**Time Spent Reviewing:**

4

---

> ### Author Response · Authors · 2021-08-09
> **Response to Reviewer Rpgw**
>
> ### We thank the Reviewer Rpgw for the four hours of effort and the associated constructive comments that will surely help improve our paper.
>
> 1. > **Reviewer**: How does this actually differ from RED (or RND even) in practice? Furthermore, once the network is learnt on the dataset, does it remain fixed throughout the duration of the experiment, or does it get periodically re-learnt?
>
> * **Response**: Both our method and RED are inspired by the RND in terms of constructing the prediction error. However, RED is an imitation learning approach for single-agent RL, which means RED uses the prediction error as the only reward signal. In contrast, we are modelling diversity in the regime of population training in multi-agent RL (i.e., the PSRO process) where at each iteration we aim to discover a new diverse and effective agent that can improve the performance of the whole population.   The network learnt on the dataset is **not** fixed through the experiment. Once a new policy is added, the aggregated Nash policy is changed. Therefore, it gets re-learnt at the beginning of each iteration of PSRO.
>
> 2. > **Reviewer**: Shouldn't $-i$ be assignable to a single opponent player? What value does it take in say a game with 3 or 4 players?
>
> * **Response**: As a general practice in game theory, $i$ indicates a single player and we utilize the special notion of $-i$ to encapsulate the remaining players. As a result, for a game with 3 players, $i$ means one specific player and $-i$ indicates the other two as a whole.
>
> 3. > **Reviewer**: Strictly speaking, I don't think the manuscript provides a good overview of the limitations of the proposed framework, nor of any particular weakness in the experimental setting used.
>
> * **Response**: We apologize for insufficient discussions on the limitations, and we include more discussions here.
>   * Regarding the limitations from the perspective of the proposed algorithm, one limitation is that the diversity weights $\lambda_1$ and $\lambda_2$ in our paper are manually tuned. Our methods target the non-transitivity in zero-sum games. Therefore, the weights for a game should be related to how strong the non-transitive component is. In future work, we will work towards quantifying the non-transitivity component of a given game automatically and determining the diversity weights correspondingly. Similar ideas have been tested in single-agent RL cases on learning the discount rate [2]. In addition, our methods also inherit the limitations from the framework of PSRO. The advantage of the PSRO based methods also depends on the amount of non-transitivity of the environment [1]. To be specific, if there aren’t too many cyclic policies involved in the environment (such as Rock->Paper->Scissor), the newest model generated by the naive self-play training paradigm could probably be the strongest one (a Nash Policy), which implies that a naive self-play would suffice to solve the problem. The PSRO methods will then lose their advantage under such circumstances since they need extra computation resources to maintain meta-payoffs.
>   * Regarding the limitations from the perspective of real-world application, the game dynamics could be complex and there could be a lot of randomnesses involved in the real-world games. Thus the approximated best response trained by reinforcement learning algorithms towards a certain policy could be inaccurate. In the Google Research Football experiment, we empirically save the checkpoint when the model win-rate is stable (i.e. the change of win-rate is less than 0.05 during two checks with the check frequency is 1000 model steps) or the training model steps reach an upper bound of 50000. These criteria could be potentially improved or further studied for all the PSRO based methods.
>
> 4. > **Reviewer**: I would personally enjoy a discussion at the very least on what other types of losses could be potentially steam from utilising the BD / RD decompositions.
>
> * **Response**: We offer here more details about what other types of losses can stem from BD and RD.
>
>   * Regarding RD, we demonstrate that many current approaches can be unified as convex hull enlargement:
>     * PSRO$_{rN}$ [3]: The rectified Nash modifies the original Nash objective, and it shows that the new objective will enlarge the convex hull more efficiently through the example of Rock-paper-scissors. Furthermore, it shows empirically that rectified Nash will lead to the largest area of the convex hull in the 2D embedding space.
>     * DPP-PSRO [4]: DPP-PSRO proves that the new policy maximizing the diversity regularized best response will strictly enlarge the convex hull in Proposition 9, which in other words, can be formulated as a convex hull enlargement problem.
>
>     To unify these, we proposed the direct objective to enlarge the convex hull and define **Response Diversity** as the contribution of a policy to the convex hull enlargement.
>
>   * Regarding BD, we show the newly proposed concept of trajectory diversity [5] can be derived by our formulation of BD as occupancy measure discrepancy here.
>     For a policy $\pi_i$, denote the trajectory distribution induced by $\pi_i$ as $q_{\pi_{i}}$ and the occupancy measure as $\rho_{\pi_{i}}$. Then trajectory diversity for $(\pi_1, \dots, \pi_n)$ is defined as the generalized JS divergence over the trajectory distributions:
>     $$
>     Diversity(\pi_1, \dots, \pi_n)=\text{JSD}(q_{\pi_1}, \dots, q_{\pi_n}).
>     $$
>
>
>     Considering that:
>     $$
>     \text{JSD}(q_{\pi_1}, \dots, q_{\pi_n})=\frac{1}{n}\sum_{i=1}^{n}D_{KL}(q_{\pi_i}||q_{\hat{\pi}}),
>     $$
>     where $q_{\hat{\pi}}=\frac{1}{n}\sum_{i=1}^{n}q_{\pi_i}$. Then following the Theorem 1 of [6], we get:
>     $$
>     D_{KL}(q_{\pi_i}||q_{\hat{\pi}})\ge D_{KL}(\rho_{\pi_i}||\rho_{\hat{\pi}}).
>     $$
>     Now we can conclude the following lower bound of trajectory diversity:
>     $$
>     \text{JSD}(q_{\pi_1}, \dots, q_{\pi_n})\ge\text{JSD}(\rho_{\pi_1}, \dots, \rho_{\pi_n}).
>     $$
>     Finally, we justify that the trajectory diversity is lower bounded by our occupancy measure level behavioral diversity. Therefore, maximizing trajectory diversity in [5] can be replaced by maximizing the lower bound BD.
>
> Finally, we appreciate your other careful and helpful suggestions (e.g., typos and writings) and will further revise our manuscript based on them.
>
> [1] Czarnecki, Wojciech Marian, et al. "Real world games look like spinning tops." arXiv preprint arXiv:2004.09468 (2020).
>
> [2] Xu, Zhongwen, Hado van Hasselt, and David Silver. "Meta-gradient reinforcement learning." arXiv preprint arXiv:1805.09801 (2018).
>
> [3] Balduzzi, David, Marta Garnelo, Yoram Bachrach, Wojciech Czarnecki, Julien Perolat, Max Jaderberg, and Thore Graepel. "Open-ended learning in symmetric zero-sum games." In International Conference on Machine Learning, pp. 434-443. PMLR, 2019.
>
> [4] Perez-Nieves, N., Yang, Y., Slumbers, O., Mguni, D. H., Wen, Y., & Wang, J. (2021, July). Modelling Behavioural Diversity for Learning in Open-Ended Games. In International Conference on Machine Learning (pp. 8514-8524). PMLR.
>
> [5] Lupu, A., Cui, B., Hu, H., & Foerster, J. (2021, July). Trajectory diversity for zero-shot coordination. In International Conference on Machine Learning (pp. 7204-7213). PMLR.

---

> > ### Comment · Reviewer_Rpgw · 2021-08-27
> > **Thank you for your response**
> >
> > Thank you very much for the detailed response. I am very happy with the feedback provided by you and the conversations with the rest of the reviewers, so I will wholeheartedly recommend acceptance.
> >
> > Looking forward to seeing the improved version at the conference!

---

### Official Review · Reviewer_qQqw · 2021-07-20

**Rating:** 6
**Confidence:** 3

**Summary:**

The aim of this paper is to train diverse populations of agents in the open-ended learning setting. To this end the authors introduce two loss terms that encourage behavioural diversity as well as response diversity. In addition they define a new evaluation metric for the performance of agent populations they call population effectivity. They carry out experiments on a synthetic non-transitive environment, the AlphaStar game as well as the Google Research Football game.


**Ethical Concerns:**

There are no ethical concerns about this work from my side.

**Limitations And Societal Impact:**

The Authors discussed this appropriately.

**Main Review:**

Positives:
* Overall the paper is well structured and not too hard to follow.
* The two main contributions (the augmented training objective and the effectivity metric) are interesting and novel in this setup as far as I can tell.
* The experiments include relevant baselines and span a good range from synthetic to more real-world. Most of the results look promising.

Concerns:
* There are parts that lack more detail. For example, the setup of the AlphaStar game is not clear to me at all. From what I understand the authors have not trained AlphaStar from scratch but rather are using payoff matrices from pre-trained agents? This needs to be more clear in order to be able to judge the impact of the results. Another example of this is the matrix for the non-transitive mixture game should be described a bit more in the main text, to at least get an intuition about it’s structure beyond having been “delicately designed”.
* I am not sure about the results on the non-transitive mixture game. I am surprised by the performance of PSRO and in particular PSRO-rN which is aimed at finding diverse strategies. I might be misunderstanding the game (and, if so, this adds to my previous point about this intuition having to be communicated better to the reader), but would the modes of the Gaussians not be where we would want the trajectories to end? If this is the case why are none of the algorithms reaching them? I am also surprised there is no cycling in the trajectories, given the cyclic nature of the game.
* A lot of the paper is in the appendix. I am aware that there is a lot to include and there is a page limit, but I would make an effort to exclude anything that is less relevant from the main text and try to bring at least intuitions about what is specified in the appendix rather than just referring to it.

Suggestions:
* The first sentence of the paper “zero-sum games involve non-transitivity” is not correct. A zero-sum game with two strategies for example is not non-transitive. You can also take any transitive game P and P-P.T will still be transitive as well as zero-sum. I am assuming that the authors might not have meant it as a universal statement, in which case they should be careful when making general statements like this and this particular sentence should be changed (if indeed they did mean it as a universal statement, this could be a concerning indication of lack of fundamental knowledge about game theory).
* Related research that might be of interest in this area is a paper on diversity of populations [1].
* Some variables are not defined, like c and W in line 120. Even if this is notation that is defined in another paper, if the authors use an equation all variables should be explained.
* In algorithm 1 the star of theta in line 5 and in line 7 is not the same, use the same one as they refer to the same thing and otherwise this could be misinterpreted.
* In lines 200 and 208 write out ‘Behavioural diversity’ and ‘Response diversity’ (you can leave out the ‘Discussions on’ part).
* There are several spelling and grammatical errors throughout the paper, I would recommend giving it a thorough scan. Here are some examples: l.24 it’s zero-sum, not zero-sums, l.133 if \rho is *a* valid, l. 214 for practical implementation, l.269 ‘condict *an* ablation study.

[1] Garnelo, Marta, et al. "Pick Your Battles: Interaction Graphs as Population-Level Objectives for Strategic Diversity." Proceedings of the 20th International Conference on Autonomous Agents and MultiAgent Systems. 2021.

**Time Spent Reviewing:**

3 hours.

---

> ### Author Response · Authors · 2021-08-09
> **Response to Reviewer qQqw**
>
> ### We thank Reviewer qQqw for his/her 3-hour efforts in offering constructive comments that will surely help improve our paper.
>
> 1. > **Reviewer**: There are parts that lack more detail. For example, the setup of the AlphaStar game is not clear to me at all. Another example of this is the matrix for the non-transitive mixture game should be described a bit more in the main text, to at least get an intuition about it’s structure beyond having been “delicately designed”
>
> * **Response**: We apologize for the lack of clarity. Regarding the AlphaStar game, your understanding is correct. We do not train AlphaStar from scratch; instead, we test our algorithm on the **meta-game** induced by 888 policies (i.e. agents) generated during the training process of solving AlphaStar, which is provided by [3]. Regarding the non-transitive mixture game, the explicit construction of the “delicately designed” payoff $\mathbf{S}$ is given in Appendix D.1. The intuition behind the construction is to ensure that when the opponent takes a pure strategy best response, which corresponds to reaching the center of one of the gaussian humps, the best response against it will be choosing among the rest gaussian humps on other directions. As a result, this game involves both strong transitive and non-transitive structures. To achieve low exploitability, an effective population has to demonstrate diverse explorative trajectories that cover all directions (see Figure 2).
>
> 2. > **Reviewer**: I am not sure about the results on the non-transitive mixture game. Would the modes of the Gaussians not be where we would want the trajectories to end? If this is the case why are none of the algorithms (e.g. PSRO, PSRO-rN) reaching them? I am also surprised there is no cycling in the trajectories, given the cyclic nature of the game.
>
> * **Response**: The reviewer is correct in the sense that the players must learn to stay close to the Gaussian centroids whilst also exploring all seven Gaussians to avoid being exploited. The reason why they (i.e. PSRO, PSRO-rN) are not reaching them is that they adopt the **approximated** best response during each iteration via gradient descent. Therefore, it is reasonable that players approached (but not fully reached) the exact best response. The failure of PSRO on such a task is because it does not use the pipeline trick. It is also reported in Figure 3 in [5]. In terms of PSRO-rN, it came as no surprise that PSRO-rN would fail in such tasks, which is also studied theoretically in Proposition 3.1 in [4] and empirically in [6].
>
>   Regarding no cycling trajectories given the cyclic nature of the game, approaching a center of a Gaussian is actually caused by the transitive component of this game. The **cyclic** component is revealed by the fact that the policies try to explore different directions.
>
>
> 3. > **Reviewer**: The first sentence of the paper “zero-sum games involve non-transitivity” is not correct.
>
> * **Response**: We agree with the Reviewer that not all zero-sum games have non-transitive components. We would like to accept your suggestion and correct it as “many zero-sum games have a strong non-transitive **component**”. This is validated by [2], which proves that a game can be generally decomposed into the transitive component and the non-transitive component. We understand the non-transitive component can be zero at many times like what you have justified.
>
> 4. > **Reviewer**: Related research that might be of interest in this area is a paper on diversity of populations [1].
>
> * **Response**: Thanks for pointing it out and we appreciate this paper. At a high level, we could expect the idea of [1] lies in the region of Response Diversity in our paper. It works towards using the interaction graphs as a more general objective to replace Nash or rectified Nash. We will include this reference in our manuscript.
>
> Finally, we appreciate your other careful and helpful suggestions (e.g., typos and writings) and will further revise our manuscript based on them.
>
> [1] Garnelo, Marta, et al. "Pick Your Battles: Interaction Graphs as Population-Level Objectives for Strategic Diversity." Proceedings of the 20th International Conference on Autonomous Agents and MultiAgent Systems. 2021.
>
> [2] Balduzzi, David, et al. "Re-evaluating evaluation." arXiv preprint arXiv:1806.02643 (2018).
>
> [3] Czarnecki, Wojciech Marian, et al. "Real world games look like spinning tops." arXiv preprint arXiv:2004.09468 (2020).
>
> [4] McAleer, S., Lanier, J., Fox, R., & Baldi, P. (2020). Pipeline psro: A scalable approach for finding approximate nash equilibria in large games. NeurIPS 2020
>
> [5] Feng, Xidong, Oliver Slumbers, Yaodong Yang, Ziyu Wan, Bo Liu, Stephen McAleer, Ying Wen, and Jun Wang. "Discovering Multi-Agent Auto-Curricula in Two-Player Zero-Sum Games." arXiv preprint arXiv:2106.02745 (2021).
>
> [6] Perez-Nieves, N., Yang, Y., Slumbers, O., Mguni, D. H., Wen, Y., & Wang, J. (2021, July). Modelling Behavioural Diversity for Learning in Open-Ended Games. In International Conference on Machine Learning (pp. 8514-8524). PMLR.

---

> > ### Comment · Reviewer_qQqw · 2021-08-17
> > **Thanks for the clarifications**
> >
> > Thanks for the response addressing some of my concerns. I think the paper would benefit from adding some of the clarifications of the rebuttal but otherwise I am happy for it to be accepted.

---

### Decision · Program_Chairs · 2021-09-27

**Decision:**

Accept (Poster)

**Comment:**

This paper explores different methods for measuring population diversity and how they can be applied in open-ended learning. All reviewers agreed that it provides a valuable contribution. One important point that emerged from the discussion with reviewer QZCS was that the paper's title is a bit of an overclaim. The authors graciously agreed to qualify it by changing to "Towards unifying...".